# Long-term balancing selection for pathogen resistance maintains trans-species polymorphisms in a planktonic crustacean

Luca Cornetti[1,2], Peter D. Fields [1], Louis Du Pasquier[1] & Dieter Ebert [1]✉

Balancing selection is an evolutionary process that maintains genetic polymorphisms at selected loci and strongly reduces the likelihood of allele fixation. When allelic polymorphisms that predate speciation events are maintained independently in the resulting lineages, a pattern of trans-species polymorphisms may occur. Trans-species polymorphisms have been identified for loci related to mating systems and the MHC, but they are generally rare. Trans-species polymorphisms in disease loci are believed to be a consequence of long-term host-parasite coevolution by balancing selection, the so-called Red Queen dynamics. Here we scan the genomes of three crustaceans with a divergence of over 15 million years and identify 11 genes containing identical-by-descent trans-species polymorphisms with the same polymorphisms in all three species. Four of these genes display molecular footprints of balancing selection and have a function related to immunity. Three of them are located in or close to loci involved in resistance to a virulent bacterial pathogen, *Pasteuria*, with which the *Daphnia* host is known to coevolve. This provides rare evidence of trans-species polymorphisms for loci known to be functionally relevant in interactions with a widespread and highly specific parasite. These findings support the theory that specific antagonistic coevolution is able to maintain genetic diversity over millions of years.

Balancing selection is a powerful mechanism to maintain genetic variation at selected loci, favoring rare alleles and disfavoring common ones[1,2]. Therefore, rare alleles are much less likely to go extinct than alleles not under selection. Balancing selection is responsible for the high level of polymorphisms observed at disease-related genes, like MHC in jawed vertebrates and R-genes in plants[3–5], and at genes involved in protein–protein interaction, like Self-Incompatibility loci in plants and mating-type loci in fungi[1]. Loci under balancing selection display a genomic footprint that includes high, sometimes extraordinarily high, genetic variation around the target of selection and an excess of genetic polymorphisms segregating at intermediate frequencies (e.g. refs. 6–9). Of particular interest is the fact that some of these polymorphisms are shared by closely related species, suggesting

that they have been maintained over time periods longer than the time since speciation occurred, and much longer than neutral polymorphisms could be expected to persist[10]. Indeed, some of these polymorphisms are independently maintained in reproductively isolated lineages that diverged millions of years ago[11]. Understanding such ancient polymorphisms opens a window into the past, helping us to see how selection acted many years ago. Evidence for such rare ancestral polymorphisms has been reported mostly for genes related to immune function, which is consistent with the idea put forth in the Red Queen model that host–parasite coevolution is a major force driving long-term balancing selection[12]. The Red Queen model of continuous host–parasite coevolution is named after a character of the same name from Lewis Carroll's *Alice in Wonderland*, who needs to run

[1]Department of Environmental Sciences, Zoology, University of Basel, Basel, Switzerland. [2]Present address: Syngenta Crop Protection AG, Stein, Switzerland. ✉e-mail: dieter.ebert@unibas.ch

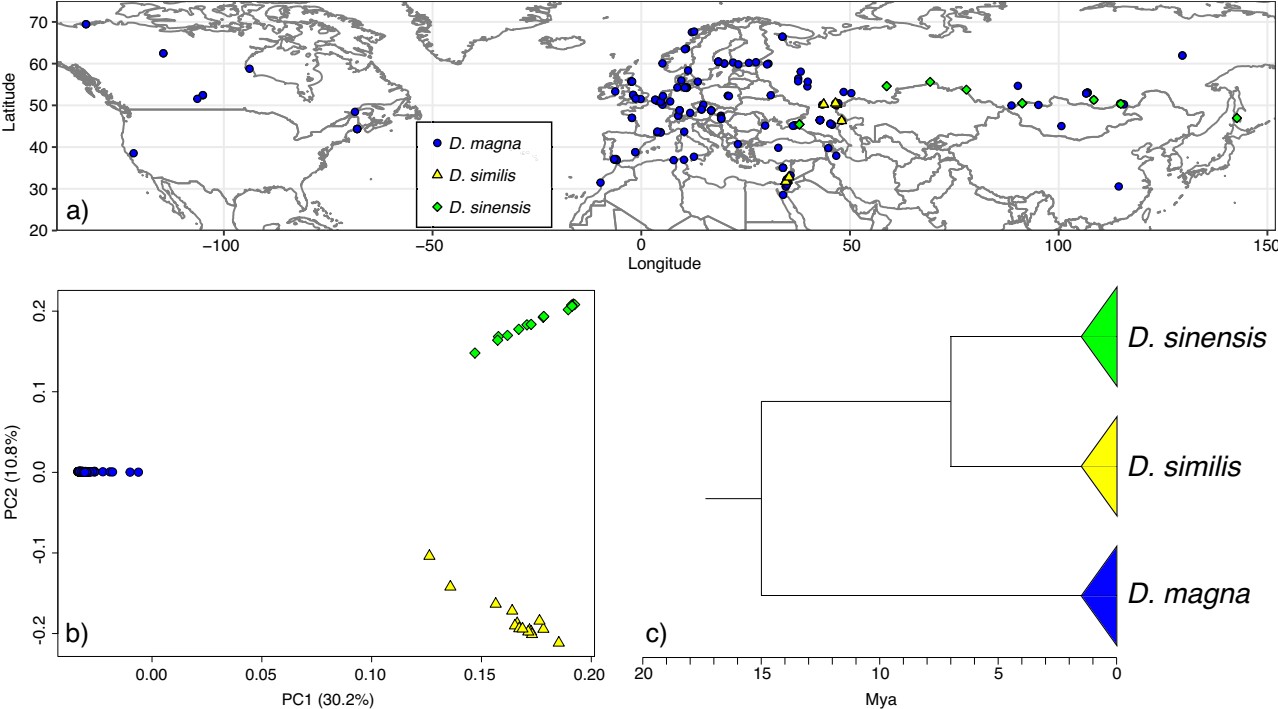

**Fig. 1 | Sampling sites, principal component analysis, and phylogenetic relations of the samples of the three *Daphnia* species used in this study. a** Map of the Northern hemisphere (20–70°N) with the *Daphnia* genotype sampling sites used in this study. **b** Principal component analysis (PCA) based on 92608 unlinked SNPs describing species diversification. **c** Simplified representation of the phylogenomic relationship among *D. magna*, *D. similis*, and *D. sinensis* from ref. 35. This tree is fossil-calibrated and based on 636 single-copy orthologs analyzed in ref. 35. Divergence times approximate the minimum estimates obtained when the most recent common ancestor of all Anomopoda (a sub-order including *Daphnia*) was dated in the Late Jurassic. Source data are provided as a source data file.

constantly to maintain her position[13]. In evolutionary theory, the idea is that parasite lines with host-genotype-specific infection profiles will be most successful if they adapt primarily to common host genotypes. These common host genotypes will then be selected against, consequently declining in frequency along with the parasites that specialized on them, giving rare genotypes a selective advantage[14–17]. The Red Queen model of coevolution is thought to underlie the high level of genetic polymorphisms at loci associated with host defense, as observed in diverse natural systems[5,6,18,19]. To date, most studies on Red Queen coevolution have focused on the host, while the long-term coevolving parasites have remained unknown.

A system with strong evidence of Red Queen coevolution is the planktonic crustacean *Daphnia magna* and its obligate and highly virulent bacterial pathogen *Pasteuria ramosa*. In this system, coevolution has been associated with high diversity at disease loci[20–22], with resistance being highly variable within and between host populations[21–23]. The same is true for parasite infectivity[20,24]. Genomic regions containing resistance loci in hosts and infectivity loci in the parasites show strongly elevated diversity levels compared to the genomic background[22]. Variation in hosts and parasites is linked by high specificity, such that infection can only result when host and parasite genotypes match[22,25,26], an important assumption of the Red Queen model and long-term balancing selection[27,28]. The *Daphnia–P. ramosa* system represents a unique model to test for footprints of long-term balancing selection in the host genome. In a Eurasian sample of *D. magna* genomes, a group of *P. ramosa* resistance genes forming a cluster was identified, at which balancing selection could be inferred from the observed patterns of genomic variation[21]. The coalescence time of the haplotypes at this cluster is greater than average for the genome. When coalescent times exceed the time since two species diverged, these polymorphisms are believed to have existed in both species since the time before the species formed, a pattern known as trans-species polymorphisms (TSP)[11]. TSP is generally rare, but has

been seen in some disease-related genes in plant and vertebrate hosts[6,29–33], but the long-term coevolving parasites are unknown. TSP represents the strongest evidence of long-term balancing selection[2,34].

In this work, we scan the genomes of three related host species for patterns of TSPs and test whether TSPs are observed in or near loci associated with polymorphisms for resistance to the virulent and widespread *Pasteuria* parasite. As part of the process, we analyze resistance phenotypes and whole genomes of 186 naturally collected *Daphnia* clones. One hundred fifty-seven clones belong to our focal species, *D. magna*, while 15 and 14 clones to two species, *D. similis* and *D. sinensis*, that together form a sister taxon to our focal species. *D. similis* and *D. sinensis* diverged about 7 Mya, while they share the most recent common ancestor with *D. magna* more than 15 Mya[35]. Our work provides rare evidence of TSPs for loci known to be functionally relevant in interactions with a widespread and highly specific parasite. Thus, our study is consistent with the theoretical prediction that specific antagonistic coevolution is able to maintain genetic diversity over millions of years.

## Results and discussion
### Resistance profiles of *Daphnia* clones
*Daphnia* clones collected from the entire Holarctic (Fig. 1a and Supplementary Data 1) were tested for resistance to five strains of *P. ramosa* isolated from *D. magna* populations across Europe. Parasite attachment to the host gut is the crucial step in the infection process and correlates strongly with infection success[36,37]. Where parasites fail to attach, the hosts are resistant. Thus, we used the parasite attachment test to determine resistance. This test is conducted with fluorescent parasite spores that can be observed through the transparent *Daphnia* bodies adhering to the cuticle of the fore- or hindgut wall of susceptible hosts, but not of resistant hosts[36,38]. All three species showed variation among clones for parasite attachment, with 33% of all *D. magna* clones testing attachment positive (susceptibility), 12.4% for *D. similis*, and

**Table 1 | Percent of positive attachment tests for clones of *D. magna*, *D. similis*, and *D. sinensis* to five *P. ramosa* isolates**

| Daphnia species | *D. magna* clade | Number clones | *P. ramosa* isolate and site of attachment | | | | | | | |
|---|---|---|---|---|---|---|---|---|---|---|
| | | | C1-foregut | C19-foregut | P15-hindgut | P15-foregut | P20-foregut | P21-hindgut | P21-foregut | All isolates and sites |
| *D. magna* (total, *n* = 157) | Western Eurasia | 125 | 33.60 | 29.60 | 64.00 | 5.60 | 36.00 | 76.00 | 3.20 | 35.43 |
| | East Asia | 18 | 11.11 | 16.67 | 50.00 | 0.00 | 11.11 | 55.56 | 0.00 | 20.64 |
| | North America | 14 | 21.43 | 21.43 | 0.00 | 35.71 | 14.29 | 64.29 | 35.71 | 27.55 |
| *D. similis* | | 15 | 6.67 | 0.00 | 6.67 | 13.33 | 20.00 | 26.67 | 13.33 | 12.38 |
| *D. sinensis* | | 14 | 28.57 | 14.29 | 78.57 | 57.14 | 28.57 | 64.29 | 57.14 | 46.94 |
| | Average: | | 20.28 | 16.4 | 39.85 | 22.36 | 21.99 | 57.36 | 21.88 | 28.59 |

*P. ramosa* isolates C1, C19, and P20 attach only to the host's foregut, while P15 and P21 are able to attach to the host's fore- and hindgut.

46.9% for *D. sinensis* (Table 1 and Supplementary Datas 1 and 2). As previously reported[36,39,40], all *D. magna* clones became infected after successful attachment of at least one *P. ramosa* isolate, while only about 64% and 74% of *D. similis* and *D. sinensis* clones, respectively, became infected after attachment (Supplementary Data 2). These reduced infection rates for *D. similis* and *D. sinensis* likely stem from the fact that our five *P. ramosa* isolates were isolated from *D. magna* and may show some degree of specific adaptation to this species[8,21]. Nevertheless, infected *Daphnia* of all three species displays the same, typical signs of the disease: the host body becomes darkish and non-transparent, and the bacterium castrates the host (Fig. 2)[41] and by this, strongly impacts host fitness. Taken together, these results show that the three *Daphnia* species suffer in the same way from *P. ramosa* infections and exhibit substantial genetic variation for resistance to *P. ramosa* that has, presumably, been maintained over a long time.

**The three *Daphnia* species show no evidence of introgression**
Next, we studied the genomes of all three *Daphnia* species to test for TSPs. First, we ruled out the horizontal transfer of genes by hybridization and introgression, which can mimic TSPs[34], by using whole genome resequencing data for all the *Daphnia* clones to assess the species' relationships, their genetic structure, and any evidence of admixture among them. A PCA approach showed high genetic differentiation among the three species, with the first PC separating *D. magna* from the other two species, which were clearly separated by the second PC (Fig. 1b and see also Supplementary Fig. 1). As previous population genomic studies have suggested[21,42], differentiation exists because *Daphnia magna* inhabits most of the Holarctic, population. The PCA of the *D. magna* clones suggests the presence of three main clusters—a Western Eurasian (WE), an Eastern Asian (EA), and a Northern American (NA) cluster, which was included in the analyses (Supplementary Figs. 2–5). A mitogenome phylogeny supported the same population and species differentiation with five lineages (Supplementary Fig. 6). In addition to the PCA analysis, which indicates a deep split among the three sampled species, we estimated an ABBA-BABA statistic (Patterson's *D*[43];) to detect gene flow/introgression among our samples. The average value of *D* for our focal species *D. magna* in relation to *D. similis* and *D. sinensis* was 0.049 and not significantly different from zero (*p* value > 0.05), indicating no evidence of recent gene flow.

**Genomic analysis of shared polymorphisms**
For each of the five *Daphnia* lineages (*D. similis, D. sinensis, D. magna* WE, *D. magna* EA, and *D. magna* NA), we performed single nucleotide polymorphisms (SNPs) calling (Supplementary Table 1), overlapped the five lists of SNPs and obtained a total of 224 high-quality, biallelic, shared polymorphisms (Supplementary Data 3). More than 10% (32 of 224) of these SNPs fall into a large region of about 250 kb in contig 11F, a region previously identified as being highly variable[39,40] and under-

balancing selection in *D. magna*[21]. The 224 SNP candidate list included only polymorphisms observed in all five clades, excluding polymorphisms in *D. magna* related to clade-specific SNPs.

Balanced polymorphisms are expected to be surrounded by one or more neutral shared polymorphisms that are in strong linkage disequilibrium (LD, i.e., less than 1 kb from each other). Therefore, we applied a filter that required at least two shared SNPs in LD[1,30], thereby identifying 35 genomic regions, including 131 SNPs, that satisfied this LD condition (Supplementary Data 4) and that we then considered for further analyses. As the further focus of our study is on the species level, we pooled the clones of the three *D. magna* lineages into one group to focus on species-level comparison in all following analyses.

To filter out from our list of shared SNPs those variants that are identical-by-state as a result of recurrent mutations, we produced allelic trees. A specific feature associated with the molecular signature of long-term balancing selection is that haplotypes from different species cluster by allele rather than by species. Thus, allelic trees do not resemble the species tree. We built trees based on fragments of variable length (i.e., from 100 bp to 1500 bp) surrounding, and centered on, each of the 131 candidate-shared SNPs. We excluded the focal (shared) SNPs from the fragments used in this analysis, as they were already a filter criterium in a previous step. For each tree, we estimated its probability of having haplotypes clustering by allele ($P_{allelic}$) following the method described by Teixeira et al. [31]. Among the 131 SNPs for which the local trees were built, we identified 31 SNPs with probabilities higher than 85 in at least one of the generated trees and considered them putative TSPs (Supplementary Datas 4 and 5). These candidate TSPs were located in 11 regions with at least two shared SNPs in LD. In these 11 regions with at least one putative TSP (in total 31 SNPs), there were an additional 25 shared SNPs (a total of 56 shared SNPs, Supplementary Data 5).

Since long-term balancing selection is expected to maintain adaptive genetic diversity, we hypothesize that TSPs fall in coding regions that show excess polymorphism. For this reason, we calculated the ratio of polymorphism to divergence (PtoD) following Teixeira et al. [31] for all annotated genes in the *D. magna* reference genome, correcting for gene length. PtoD, which accounts for mutation rate, was obtained as $p/(d+1)$ where $p$ is the number of polymorphisms in the samples of one species and $d$ is the number of fixed SNPs between species. We then focused on the genes containing putative TSPs and calculated where the candidate genes fell relative to the empirical PtoD distribution of all genes. Each of the 11 regions with at least two shared SNPs in LD contained one gene with putative TSPs. Four of these genes showed exceptional variability, with the average of the four PtoD percentiles falling in the top 10% of PtoD distribution and no single value over the 15% percentile (Table 2 and Supplementary Data 6). Finally, the topologies of the gene trees built in the surrounding of the four remaining candidates suggest that haplotypes indeed cluster by allele and not by species (Fig. 3 and Supplementary Fig. 6), providing strong support for TSP.

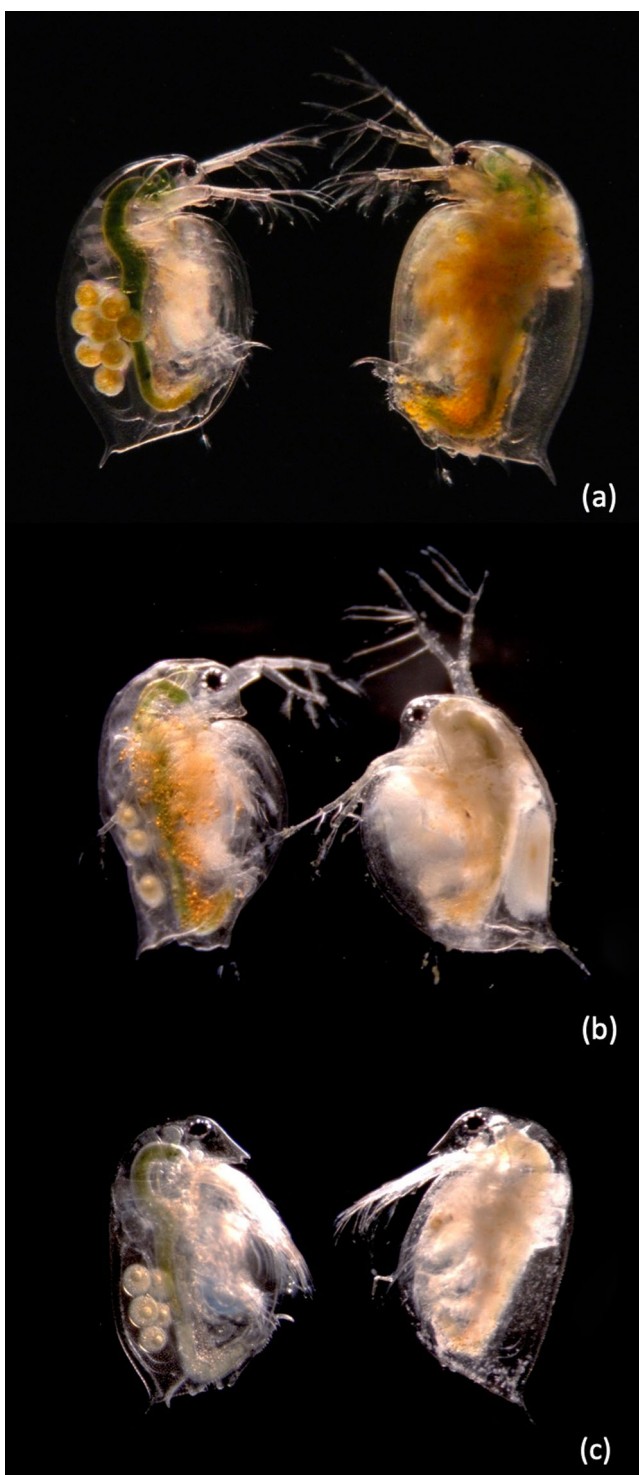

**Fig. 2 | Photographs of *P. ramosa* infected and healthy females of the three *Daphnia* species used here.** *D. magna* (**a**), *D. similis* (**b**), and *D. sinensis* (**c**) phenotypes of infection. On the right are animals infected with *P. ramosa*, while on the left are healthy animals. The body of infected hosts becomes non-transparent. The bacterium castrates the host, recognizable by the absence of eggs in the brood chamber.

## Population genetic analysis of shared polymorphisms

Balancing selection maintains polymorphisms at selected loci, resulting in characteristic genomic signatures[1,10] such as elevated nucleotide diversity ($\pi$) and Tajima's *D*, which are indicative of increased genetic variation around the target of selection and excess polymorphisms segregating at intermediate frequencies, respectively. Balancing selection also reduces population or species differentiation as measured by $F_{st}$[1,10]. We therefore calculated $\pi$, Tajima's *D*, and $F_{st}$ for non-overlapping sliding windows of different sizes (i.e., 1 kb, 2.5 kb, and 5 kb) in the regions surrounding our candidate TSPs. The same was done for the 20 longest contigs and the contig where the TSP is located that functioned as background level. The target region was identified as a 100 kb genomic segment centered on a candidate TSP. For each species, we then assessed whether $\pi$ and Tajima's *D* were elevated and if $F_{st}$ values were lower in the candidate regions compared to the background level, as theoretically expected for sites under long-term balancing selection (Supplementary Figs. 7–15 and Supplementary Tables 2–10). For our candidates, regardless of the size of the sliding window applied, we found a significant deviation from the background level in most of the statistics indicative of balancing selection (Table 2). These observations suggest that in each of the three reproductively isolated and highly divergent *Daphnia* species, the genomic regions around the putative TSPs display high genetic variability and that the species are less differentiated in these regions than expected from a genome-wide average. We further validated these results by applying the normalized *ß* statistics, a test specifically designed to detect clusters of polymorphisms segregating at similar frequencies, a pattern associated with signatures of long-term balancing selection[44]. Most comparisons showed significantly elevated *ß* scores in the target regions compared to the background level (Table 2, Supplementary Figs. 16–18, and Supplementary Tables 11–19), further supporting our hypothesis that the regions surrounding TSPs evolve under long-term balancing selection.

To ensure that our observations were not biased by the occurrence of paralogs, which can cause read mismapping, or by other artifacts, we Sanger-sequenced the flanking regions of the identified TSPs. We included this control step because sequence duplication is known to determine pseudo-heterozygosity[45]. In addition, we wanted to ensure that there were no inconsistencies between the genotypes obtained with Illumina sequencing and the ones obtained with Sanger sequencing (details in Supplementary Tables 20). We were able to confirm the Illumina genotypes and the presence of a plausible number of heterozygous positions surrounding all our candidates (examples of chromatograms in Supplementary Figs. 19–24).

## Functional analysis of the shared polymorphisms

The four candidate regions revealed interesting associations. Two of them are located on contig 11F, in close association with a large cluster of resistance loci for the virulent parasite *P. ramosa*. This cluster includes four known resistance genes, A, B, C, and F[39,40], and has been shown to be under balancing selection in a Euasian *D. magna* sample[21]. A, B, and C form a cluster with strongly reduced recombination, while the F locus is directly adjacent to it. One of the TSPs is situated about 48 kb before the ABC cluster, while the other is located within the F locus. The TSP observed on contig 18F is located in a region rich in immunity-related genes (e.g., multiple lectins, TRIMs, clip serine protease), about 101 kb before another *Pasteuria* resistance locus (the D locus)[38], while contig 28 F includes multiple genes with immune functions (e.g., lectins, clip serine proteases, complement related elements). These observations suggest that long-term balancing selection maintains, independently in three *Daphnia* species, ancestral alleles in genomic regions that have a functional link with *P. ramosa* resistance

Our best TSP candidates, three of which showed at least a non-synonymous TSP, can be classified as immune-related genes or genes involved in immunity pathways (Table 2). The gene 11F-21.35, which is in close proximity with the ABC,F resistance loci against *P. ramosa*, encodes a C1q domain-containing protein. C1q are immune molecules responsible for initiating the complement pathway in vertebrates. Their role in invertebrate immunity has been demonstrated in mollusks and their presence is documented in crustaceans[46]. This gene was

**Table 2 | PtoD estimates and summary statistics indicating balancing selection for the four genes that included at least one putative TSP**

| Contig-transcript | TSP | Gene function | PtoD (p) value (percentile) | | | | Nucleotide diversity (π) | | | Tajima's D | | | $F_{st}$ | | | β score | | |
|---|---|---|---|---|---|---|---|---|---|---|---|---|---|---|---|---|---|---|
| | | | D. magna vs D. sinensis | D. magna vs D. similis | D. similis | D. sinensis | D. magna | D. similis | D. sinensis | D. magna | D. similis | D. sinensis | D. magna vs D. similis | D. simi-lis vs D. sinensis | D. magna vs D. sinensis | D. magna | D. similis | D. sinensis |
| 000011F-21.35 | 2,141,058 | C1q domain-containing protein | 50.7 (0) | 50.7 (0) | 39.1 (0) | 37.6 (0) | *** | *** | *** | * | ** | *** | *** | NS | *** | *** | *** | *** |
| 000011F-23.73 | 2,343,857 | Cladoceran-specific protein | 12.6 (9) | 50.2 (0) | 28.6 (0) | 12.6 (1) | *** | * | NS | ** | *** | ** | *** | ** | *** | *** | *** | *** |
| 000018F-16.99 | 1,611,159 | Alpha subunit of putative Na+/K+ ATPase | 6.2 (15) | 7.2 (14) | 4.6 (5) | 5.1 (3) | *** | *** | * | NS | NS | NS | NS | * | NS | ** | ** | ** |
| 000028F-3.33 | 298,723 | Metalloendopeptidase | 15.2 (7) | 7.6 (13) | 3.5 (6) | 14.1 (1) | NS | *** | *** | NS | NS | *** | ** | *** | ** | *** | *** | *** |

Details of the position and gene function for the four genes that included at least one putative TSP are reported. PtoD values (corrected for gene length and its value multiplied by 1000) along with their percentiles (in parenthesis) compared to the empirical PtoD distribution of all genes for each species are given. PtoD values for *D. magna* are calculated in comparison to both *D. sinensis* and *D. similis* (for more details see Table S7). Nucleotide diversity, Tajima's *D*, $F_{st}$, and β score were calculated in 5 kb non-overlapping sliding windows and we report here the statistical significance of the comparison of the candidate regions, identified as a 100 kb genomic segment centered on the candidate TSP, with the background level of genetic diversity identified as the 20 longest contigs in the genome. All tests are two-sided Wilcoxon tests. *p* Values, adjusted for false discovery rates, represent results from tests for elevated nucleotide diversity, Tajima's *D* and β score relative to the background, and for reduced $F_{st}$ (***<0.001; **<0.01; *<0.05; NS ≥ 0.05).

shown to be upregulated in the *D. magna* transcriptome 8 days and 16 days after *Pasteuria* exposure[47]. Another gene, 11F-23.73 contained in the *Pasteuria* resistance F locus, is a member of a large "Cladoceran-specific protein" family[40]. Genes in this family, 11F-23.73 included, have features suggesting they play a role in host–pathogen interaction, such as signaling capacity via a putative STAT binding site and differential expression between susceptible and resistant clones[40]. The two remaining candidate genes on contigs 18 and 28 have no known direct immune-related function. Candidate 18F-16.99 is a Na+/K+-ATPase that can function as a signal transducer/integrator to regulate the production of reactive oxygen species, which are involved in a large number of processes including macrophage-mediated immunity[48]. Gene 28F-3.33 is a metalloendopeptidase, which is considered to play a major role in regulating numerous physiological and pathological processes, as well as immune responses in invertebrates[49]. While our candidates are enriched for immune function-related genes, we need to keep in mind that also other evolutionary processes, might be responsible for the maintenance of TSPs.

In conclusion, our study provides genomic evidence of TSPs in or close to known resistance loci (ABC, D, and F loci) in the focal species of this study, *D. magna*. The loci are involved in the coevolutionary interactions between the *Daphnia* host and *P. ramosa*. In order to be effective, the selective agent underlying the maintenance of the TSPs should be geographically widespread, should be present over an extended period of time, and should show similar tropism towards related host species[50]. The parasite *P. ramosa* fulfills all these criteria: it has been shown to have a wide geographic distribution[51], a long history of coevolution with its obligate hosts[21,52,53], and—as demonstrated here—phenotypic variation in resistance and a similar infection phenotype among evolutionary-related species (Table 1 and Fig. 2). Together, these results suggest that long-term balancing selection for *P. ramosa* resistance is a determinant in the maintenance of ancestral alleles in *D. magna* and the sister clade that shared a common ancestor at least 15 million years ago. Examples of TSPs, as a signature of ancient coevolution, are thought to be extremely rare[30], and indeed, we found only a few instances of identical-by-descent TSPs in our study. In contrast to earlier reports, our study provides compelling evidence for the causal agent driving coevolution in this system, and thus, corroborates the hypothesis that specific—one host–one parasite—antagonistic coevolution can be an important factor in the generation and maintenance of genetic diversity[2]. It represents to our knowledge the first evidence of long-term balancing selection concerning a specific infectious disease and the loci directly underlying resistance to this disease.

## Methods

### Samples and phenotyping

*Daphnia* are cyclical parthenogens that can be maintained as stable clones under lab conditions in an asexual mode of reproduction. These clonal cultures produce sufficient material for whole genome resequencing. We sequenced 186 clones from three *Daphnia* species: *D. magna* (*n* = 157 clones), *D. similis* (*n* = 15), and *D. sinensis* (*n* = 14). These lines were collected and isolated from natural populations as adult females or resting eggs and maintained under laboratory conditions as clonal lines[35,42,54]. *D. similis* and *D. sinensis* are suggested to be sister species; together, they are close to *D. magna*. Another newly identified sister species of *D. magna* is only known from one population and was therefore not used here[35]. A phylogenomic analysis has suggested that *D. magna* diverged from *D. similis*/*D. sinensis* at least 15 Mya, while *D. similis* and *D. sinensis* share the most recent common ancestor about 7 Mya[35]. In our study, we aimed to include the widest geographic spread for *Daphnia* clones, which encompasses the entire Holarctic for *D. magna*, and much of Eurasia for the other two species (Fig. 1a). Details of the clones included in the study are reported in Supplementary Datas 1 and 2.

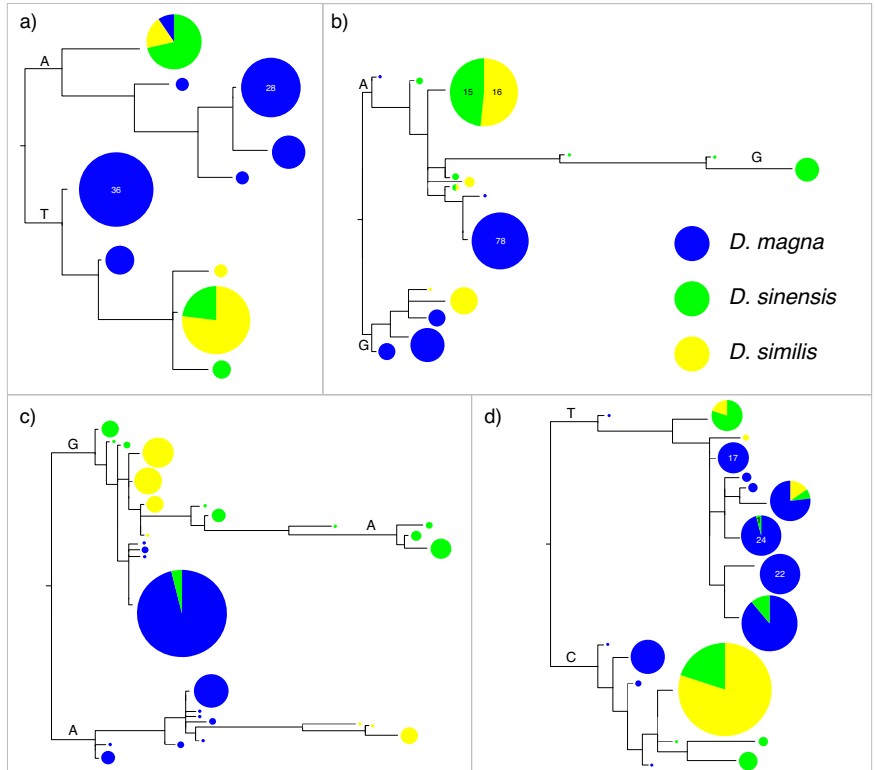

**Fig. 3 | Neighbor joining trees of haplotypes including the focal shared SNPs with evidence for TSP.** The trees based on the sequences surrounding the best TSP candidates are reported: **a** 11F-21.35–100 bp, **b** 11F-27.73–100 bp, **c** 18F-16.99–200 bp, and **d** 28F-3.33–200 bp. In these trees, haplotypes cluster by allele and not by species, i.e., each cluster contains three colors: *D. magna:* blue, *D. similis*: yellow, and *D. sinensis*: green. In some cases, external branches were collapsed in order to facilitate tree representation. The alleles reported on the branches indicate the SNP variant present in all the haplotypes in the relative cluster. The circle area is proportional to the haplotype frequency, except when the haplotype frequency is too high for the purpose of the representation and a number is reported indicating the total of similar or identical haplotypes. Source data are provided as a source data file.

We tested all *Daphnia* clones for resistance to different isolates of the bacterial parasite *P. ramosa*. For each *Daphnia* genotype, we estimated the resistance phenotype (resistotype) using five *Pasteuria* strains (C1, C19, P15, P20, and P21) isolated from natural *D. magna* populations across Europe. Since the attachment of parasite spores to the host is the most important step in the infection procedure and explains about 90% of the overall variation in infection success[36,41], resistance is defined as the ability of the host to prevent attachment to a given attachment site. The attachment test described in ref. 36 allows us to score resistotypes by exposing the host to fluorescently labeled *Pasteuria* spores and then observing whether these spores attach to the cuticle of the *Daphnia's* fore- or hindgut, where the parasite penetrates the host's cuticle and enters its body cavity[36,41]. Specifically, we assessed whether labeled spores attached to the host's foregut (all five *Pasteuria* strains) and/or hindgut (two *Pasteuria* strains: P15 and P21). For each *Daphnia* genotype, six replicates were performed, and a genotype was considered susceptible to the bacterial strain when more than half of the replicates showed attachment (Supplementary Data 1). In *D. magna*, resistance and susceptible phenotypes were found to be uniformly distributed across the entire species distribution (Supplementary Data 1) as previously shown in ref. 21, including the newly analyzed North American clones.

Attachment of *Pasteuria* spores in the native host species from which it was isolated leads to successful infection[8,36]. Attachment polymorphisms are also seen in related non-native host species; however, in these cases, the correlation between attachment and successful infection is reduced[8,36]. Besides attachment tests, we performed infection trials by experimentally exposing *D. similis* and *D. sinensis* clones, that were positive to spore attachment to the five previously mentioned *Pasteuria* strains, following the protocols of

Luijckx et al. [8]. To determine infection, we observed crushed *D. magna* host bodies under a phase contrast microscope (400×); individual *Daphnia* were considered infected when the presence of *Pasteuria* spores was detected. When at least one replicate of a clone became infected, we considered the clone infectable.

## DNA isolation and sequencing

Before genomic DNA extraction, we reduced bacterial and other non-target DNA. For this, all animals were kept for three days in a solution of Ampicillin, Streptomycin, and Tetracycline (Sigma) at a concentration of 50 mg/l each, and transferred daily into a fresh antibiotic solution. To reduce gut content, the animals were not fed during this three-day treatment, instead receiving 5 mg of superfine beads of the gel filtration resin Sephadex ® G-25 (Sigma-Aldrich) twice a day in their medium. Sephadex causes gut evacuation when ingested by the animals. We extracted DNA from 20 to 50 animals of each clone using the QIAGEN Gentra Puregene Tissue Kit, including the RNaseA (100 mg/ml; Sigma) digestion step. Whole-genome Illumina paired-end sequencing (read length 125 bp) was performed by the Genomics Facility service platform at the Department of Biosystem Science and Engineering (D-BSSE, ETH) in Basel, Switzerland, on an Illumina HiSeq 2500.

## SNP calling

Raw reads were trimmed to remove adapters using Trimmomatic 0.36[55]. We aligned the trimmed reads to a newly assembled high-quality reference genome of a three-times selfed Finnish *D. magna* clone (the same genotype for which a previous genome assembly has been released (NCBI accession LRGB00000000.1)). This reference genome, assembled using the FALCON assembler[56], was obtained

using high-coverage PacBio long-read sequencing (see Data availability). In order to increase the accuracy of read mapping, different approaches were used for the reads obtained from the different species. For *D. magna* clones, BWA v0.7.7[57], a software package for mapping low-divergent sequences against a reference genome, was used to align the reads. To align the reads from *D. similis* and *D. sinensis* clones, we used Stampy v1.0.32[58], an algorithm specifically designed to account for the divergence between the reference genome and the mapping reads. Polymorphisms were called using the HaplotypeCaller function in GATK 4.0[59]. Using VCFTOOLS 0.1.16[60], we retained only high-quality biallelic SNPs, satisfying the conditions of minimum genotype depth of 10, maximum genotype depth of 100, minimum genotype quality of 30, maximum amount of missing data of 50%, and minor allele frequency of 0.05.

## Genotype clustering and test for introgression
Working with the dataset of all 186 *Daphnia* clones, we used a Principal Component Analysis (PCA) implemented in the R package SNPRelate[61] to detect possible evidence of recent introgression among the three species. In order to account for the linkage between markers, we analyzed only SNPs at least 1 kb distant from each other. As expected, the PCA approach showed high genetic differentiation among the three species, with the first PC separating *D. magna* from *D. similis* and *D. sinensis* (Fig. 1b and Supplementary Figs. 1 and 2). Since PCA can be biased by unequal sample size[62], we repeated the PCA analysis with subsamples of only 15 *D. magna* clones. The result remained unchanged (Supplementary Fig. 1). A PCA analysis was also used to assess evidence of population structure and recent introgression events among lineages within *D. magna* that inhabit most of the Northern Hemisphere. It has already been shown that population structure exists in *D. magna*[42,63]. Here, we increased the number of *D. magna* clones and included clones from North America for the first time (Fig. 1). For this reason, a detailed assessment of population differentiation among the *D. magna* clones was necessary. The PCA results obtained for the *D. magna* clones confirmed the presence of three main clusters, namely a WE, an EA, and an NA cluster (Supplementary Fig. 4).

To test for recent introgression among the three *Daphnia* species, we estimated an ABBA-BABA statistic (or Patterson's $D$[43];), which detects geneflow among samples. We used the same genotypes as for our PCA analysis, as well as the program *Dsuite*[64] to estimate $D$. Significant elevation of $D$ above zero was assessed by applying a block-jackknife resampling procedure[65,66] with 20 subsamples.

## Mitochondrial genome assembly and mitogenome phylogeny
We assembled the mitochondrial genome for a subset of *Daphnia* clones following the protocol described in ref. 35. The mitogenomes were then annotated using the MITOS web server[67]. We aligned the protein-coding genes with MUSCLE v3.8.31[68] and concatenated them to generate a mitochondrial phylogeny using the software RAxML v8.1.20[69]. The best ML tree was obtained by assuming a general time reversal model of sequence evolution with a gamma-distributed model of rate heterogeneity, taking into account gene partitioning. In order to test for tree reliability, we performed a bootstrap approach by generating 100 pseudo-replicates. *Daphnia hispanica* was used as an outgroup (ENA accession number: LS991493), as this species is known to be closely related to the species considered here[35].

The best ML mitochondrial phylogeny resembled the previously reported topology[35] with *D. similis* and *D. sinensis* being sisters as part of the *similis* group described in ref. 70. The *similis* group, in turn, is sister to *D. magna*. The obtained tree was highly supported with almost all internal nodes, showing a bootstrap value of 100. Although their evolutionary relationship cannot be fully resolved, the three *D. magna* lineages highlighted by genome-wide polymorphisms were also highly supported by the mitogenomes (Supplementary Fig. 5). Overall, the

mitogenomes suggest, as shown by SNP data, the occurrence of five main lineages among the *Daphnia* analyzed in this study.

## Finding shared SNPs
After defining the five main lineages of the *Daphnia* clones in this study, we performed SNP calling within each of these lineages. We used VCFTOOLS 0.1.16[60] and the requisites specified above to obtain lists of high-quality polymorphisms for each of the five *Daphnia* lineages (*D. magna* from North America (13), *D. magna* from Eastern Asia (18), *D. magna* from Western Europe (126), *D. similis* (15), and *D. sinensis* (14). We then overlapped the lists of SNPs to identify shared polymorphisms among species/lineages. We obtained a total of 227 polymorphisms shared among the five lineages. In order to obtain a reliable list of shared SNPs, we ensured that the identified polymorphisms did not lie in low-complexity regions (i.e., regions rich in repeats), where were identified in our reference genomes using the software DustMasker 1.0[71]. The three identified shared SNPs that were found to occur in low-complexity regions were excluded from our set of candidate polymorphisms.

## Distinguishing between shared and TSP
In order to exclude variants from our list of shared SNPs that are identical-by-state as a result of recurrent mutations, we used, as a reference, features associated with the molecular signature of long-term balancing selection. The main feature is that haplotypes from different species cluster by allele, rather than species, i.e., they produce allelic trees, and not trees resembling the species tree. These haplotypes include polymorphisms that predate species splits, while neutral recurrent shared polymorphisms are expected to generate species trees[72]. Because of the long-term effects of recombination, we expect this signature to be restricted to a short genomic region around the putative TSP[1]. In addition, this segment is likely to contain at least two shared polymorphisms that arose in the ancestral populations of *D. magna-D. similis-D. sinensis* and are in strong or complete LD with the selected site[30]. Thus, we used these properties to exclude SNPs from our candidate list that resulted from recurrent mutations. Specifically, we required shared SNPs to be in LD (i.e., within 1 kb) with at least one other shared SNP. Among the 224 shared SNPs retained after the previous filtering (listed in Supplementary Data 3), we identified 35 regions (including 131 SNPs) having at least two shared SNPs in LD and were considered for further analyses (Supplementary Data 4).

For the polymorphisms that satisfied the above condition, we generated haplotype trees based on haplotype alignments of putative TSPs flanking regions. We selected regions of variable length (100 bp, 150 bp, 200 bp, 400 bp, 600 bp, 800 bp, 1000 bp, and 1500 bp) centered on the putative TSPs and inferred the chromosomal phase for each fragment using the software SHAPEIT4[73]. Phasing was done for all species and samples simultaneously, including the use of individual BAM files to incorporate read-based phasing. Then we built local trees excluding the putative TSPs from the sequences. For this, we used a subset of the *Daphnia* clones included in the study consisting of all the 15 *D. similis* clones, all the 14 *D. sinensis* clones, all 13 *D. magna* clones from North America, all 18 *D. magna* genotypes from Eastern Asia and 20 randomly selected *D. magna* clones from Western Eurasia. We performed this subsampling procedure so as to have similar sample sizes among species/lineages. Because our focus was at the species level in this and all subsequent analyses, we pooled the clones of all three *D. magna* lineages into one group. For each allele tree, which included 160 haplotypes, we estimated the probability of having haplotypes clustering by allele ($P_{allelic}$) following the method described in ref. 31. Specifically, for each generated tree, we applied a resampling strategy with six randomly selected haplotypes (one haplotype per allele and per species) 1000 times. With these haplotypes, we built neighbor-joining trees that were considered allelic trees if the three closest tips were haplotypes with the same allele of the three species.

In this way, for each of the local trees surrounding a putative TSP, we obtained a $P_{allelic}$ that was estimated as the proportion of trees having haplotypes clustering by allele over the 1000 resampled trees. A shared SNP was considered a candidate TSP when it showed a $P_{allelic} > 85$ in at least one of the generated trees (i.e., trees based on regions of variable length).

### Annotation of TSPs and gene function identification

The retained SNPs from the previous step were annotated and their effect was predicted using the software SnpEff[74] (Supplementary Data 5). Among those, 47 (84%) were located in coding sequences, 17 of which were non-synonymous and 28 synonymous variants. One substitution determined the gain of a stop codon (Supplementary Data 5). The 11 identified regions overlapped with 11 different genes, which are reported alongside their function in Supplementary Data 6.

Since *D. magna* genome annotation is not as accurate as the *D. pulex* annotation, we used a blast approach to obtain the *D. pulex* orthologs of the 11 genes containing TSPs in order to infer a functional classification. Specifically, we used blastp[75] and the fasta sequences of the *D. magna* proteins as queries against a *D. pulex* protein database (from: http://wfleabase.org/genome/Daphnia_pulex/dpulex_genes2017/genes/). We used an *e*-value threshold of 1*e*-40 and considered the most likely ortholog of the hit showing the lowest *e*-value. For one gene (genemark-000017F-processed-gene-17.81-mRNA-1), we did not identify sequence similarities in *D. pulex* even when the *e*-value threshold was increased to 1*e*-10. Since this suggested there is no *D. pulex* ortholog for this gene, it was therefore classified as an "uncharacterized protein" in the *D. magna* annotation. We then used Panther (v16.0)[76,77] to obtain a functional classification for each of the remaining ten genes using *Daphnia pulex*, included in the Panther database, as a reference organism. For six of the 11 genes, we could not find a defined function. These genes may represent cases of *Daphnia*-specific gene families (accounting for about 36% of *D. pulex* genes) that do not have homologs[78], or they might be the result of some drawbacks of our blast approach. All genes were further inspected with InterProScan (v5.48)[79] to provide additional validation of gene function and assess the presence of specific domains. Overall, of the 11 genes containing TSPs, two could be associated with an immune-related function (genemark-000011F-processed-gene-21.35-mRNA-1: C1q domain-containing protein, maker-000011F-snap-gene-23.73-mRNA-1: cladocera-specific protein).

### Polymorphism-to-divergence ratio

Since long-term balancing selection is expected to maintain adaptive genetic diversity, we hypothesized that TSPs fall into coding regions that show an excess of polymorphism. We thus calculated the ratio of PtoD following[31] for all genes in our reference genome. PtoD, which accounts for mutation rate, was obtained as $p/(d + 1)$ where $p$ is the number of polymorphisms in one species and $d$ is the number of fixed SNPs between species. We calculated $d$ with vcftools as the number of SNPs showing an $F_{st} = 1$ in the following species pairs: *D. magna* vs *D. sinensis* for *D. sinensis*; *D. magna* vs *D. similis* for *D. similis*; for *D. magna* we made two comparisons: *D. sinensis* vs *D. magna* and *D. similis* vs *D. magna*. We then corrected for a slightly positive correlation ($R = 0.06$, $p < 0.05$) observed between PtoD and gene length by dividing the calculated PtoD by the gene length; the obtained value was multiplied by 1000 to make it more readable. We then focused on the corrected PtoD values observed in the genes containing putative TSPs and determined where the candidate genes fell into the empirical PtoD distribution of all genes. Haplotype trees obtained in the flanking regions of the TSPs identified in our top candidate genes are shown in Fig. 3.

### Molecular signature of balancing selection in the candidate regions

A molecular signature of (long-term) balancing selection is high genetic diversity in correspondence with TSPs, since polymorphisms near these loci should be more ancient than the average genome-wide coalescent time[72]. We, therefore, calculated nucleotide diversity ($\pi$) for the regions surrounding the TSPs in non-overlapping sliding windows of 5000 bp, 2500 bp, and 1000 bp using VCFTOOLS 0.1.16[60] and assessed, within each species, if $\pi$ was elevated compared to the background level of genetic diversity (i.e., the contig where the TSP is located, excluding the candidate region, and the 20 longest contigs in our reference genome without loci known to be associated with *Pasteuria* resistance). We used multiple sliding window sizes in order to test if the size of the window influences the results, given that balancing selection is expected to leave its molecular footprints in relatively short genomic fragments[1]. Furthermore, because the allele frequency distribution for sites under balancing selection is expected to exhibit a trend towards intermediate frequencies, we tested this hypothesis with the neutrality test Tajima's $D$[80], which summarizes polymorphism data based on the difference between the average pairwise diversity ($k$) and the number of segregating sites ($S$). When there is an excess of polymorphisms at intermediate frequency, $k$ increases more than $S$, resulting in a positive Tajima's $D$. Within each lineage, we also assessed, using the same approach as above (i.e., calculating Tajima's $D$ in non-overlapping sliding windows of 5000 bp, 2500 bp, and 1000 bp), if Tajima's $D$ displayed significantly elevated values in the regions surrounding TSPs compared to the background level. Moreover, theory predicts that balancing selection reduces population differentiation, as measured by $F_{st}$[1,30]; consequently, $F_{st}$ values lower than the background level, are expected in the proximity of ancestral polymorphisms. We tested this prediction by comparing $F_{st}$ obtained in non-overlapping sliding windows of 5000 bp, 2500 bp, and 1000 bp in the candidate regions vs. the background.

In order to validate these candidate regions, we also performed a test specifically designed to detect signatures of long-term balancing selection[44]. We calculated, using the Betascan program[44], the $\beta$ score that quantifies allele frequency correlations between SNPs. We specified the length of the window surrounding each SNP following the formula in ref. 44. For window size calculation we used *Daphnia*-specific parameters ($T$, time since balancing selection, and $\rho$, the recombination rate) as in ref. 21, which resulted in a window size of 250 bp (option-w 250). For the other population genetic indexes, we averaged $\beta$ scores in non-overlapping sliding windows of 5000 bp, 2500 bp, and 1000 bp that were then compared between candidate regions and backgrounds. These analyses were repeated three times using different minor allele frequency thresholds (i.e., 0.05, 0.1, and 0.17, following[44]) to explore the effects of this parameter on the results. The results proved to be very consistent (Supplementary Tables 11–19), and in the main text, as well as in Table 2, we report the $\beta$ score results obtained with a 5000 bp sliding window and a minor allele frequency of 0.05.

Contig 11F, where two genes including TSPs were identified and where the ABC and the F *Pasteuria* resistant loci are located[39], is about 3 Mb long. The ABC locus is characterized by a large (i.e., about 150 kb) nonhomologous region (NHR) segregating in susceptible and resistant *D. magna* clones[39]. Therefore, we excluded the polymorphisms in this NHR in our comparisons. We plotted the obtained results and performed statistical analyses using Wilcoxon rank sum test) with ggplot2 and *R*.

### Functional annotation

Our study highlighted four genes as the most likely candidates to have evolved under long-term balancing selection and to contain TSPs. However, as we did not obtain a precise functional annotation using a blast approach for one of the four genes, we initially classified it as an "uncharacterized protein" (Supplementary Data 6). In fact, this "uncharacterized protein" (gene 11F.23.73) had already been identified as a member of a very large immunity-related gene family (the Cladoceran-specific protein" family with over 100 members) and was also an outlier

in a study that focused on the *P. ramosa* resistant locus *F*[40]. This gene, 11F-21.35, has several features suggesting involvement in the interaction between *Daphnia* and *Pasteuria*. It encodes a protein with two domains: a COLFI domain (the fibrillar collagen C-terminal domain) and a globular C1q domain at the C terminus. Both domains are known to have binding capacities. The members of the C1q family can be involved in host defense, inflammation, apoptosis, and autoimmunity[81]. As a pattern recognition molecule, C1q can engage a variety of ligands via its globular domain and ultimately modulate cells of the immune system. Its role in invertebrate immunity has been demonstrated in several mollusks[82], and its presence is documented in crustaceans[46].

The gene 18F-16.99 encodes a Na⁺/K⁺-ATPase. Na⁺/K⁺-ATPase can function as a signal transducer/integrator to regulate the production of reactive oxygen species.

Reactive oxygen species are involved in a large number of processes including macrophage-mediated immunity[48]. They can engage in direct antimicrobial activity against bacteria and parasites, as well as redox-regulation of immune signaling and induction of inflammasome activation.

### Reporting summary
Further information on research design is available in the Nature Portfolio Reporting Summary linked to this article.

## Data availability
The genomic data generated in this study have been deposited in the NCBI database under accession code BioProjectID PRJNA995356. The reference genome used is available at Zenodo (https://zenodo.org/records/11283641)[83] and in the NCBI database under accession code BioProject ID PRJNA624896 (https://www.ncbi.nlm.nih.gov/datasets/genome/GCA_040143795.1/). The data generated in this study are provided in the Supplementary Information/source data file. The utilized VCF is available at Zenodo (https://zenodo.org/records/11099779)[84]. Source data are provided with this paper.

## Code availability
Scripts required for replicating our results are available at Github[84] repository: https://github.com/ebertlab/transpecies_polymorphism_manuscript.

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

## Acknowledgements

We thank J. Hottinger, U. Stiefel, and M. Krebs for help in the laboratory and the Genomics Facility Basel for sequencing the samples. We thank Yan Galimov for providing *Daphnia* samples from Russia and Frida Ben-Ami for samples from Israel. We thank members of the Ebert group for their feedback on the study and the manuscript. We thank S. Zweizig for language editing. This work was supported by the Swiss National Science Foundation (SNSF) (grant numbers 310030B_166677 and 310030_188887 to D.E.).

## Author contributions

All authors conceived the study. D.E. organized and collected the samples. L.C. and P.D.F. conducted the sequencing work. L.C. conducted the bioinformatic analysis with the help of P.D.F. and D.E. L.D.P. annotated and analyzed the candidate genes. L.C. wrote the manuscript, which was read, edited, and approved by all authors.

## Competing interests

All authors declare no competing interests.
