## [Peer Review File · Nature Communications]

Long-term balancing selection for pathogen resistance maintains trans-species polymorphisms in a planktonic crustaceanReviewers' Comments:

Reviewer #1:

Remarks to the Author:

The manuscript by Cornetti et al aims to detect genomic regions under long-term balancing selection, by looking for trans-species polymorphisms shared between three closely related crustacean species (*Daphnia magna*, *D. similis* and *D. sinensis*). They identified 14 genes containing identical-by-descent SNPs, of which 5 further exhibit evidence for balancing selection, and three are near loci involved in resistance to a known pathogen (*Pasteuria ramose*).

I found the manuscript of great interest and took a lot of pleasure reading it, as it combines thorough population genomics analyses with experimental work to decipher the evolutionary dynamics of loci under antagonistic coevolution. Furthermore, the text and figures are clear and well interpreted.

Main comments:

- Could you elaborate more in the main text on why the resistance/susceptible phenotype is so variable across *Pasteuria* strains (Table S1, Methods L50-56)? Is there a significant interaction between the *Pasteuria* strain and the host species/population?
- L122-124: I don't think that observing an overall genetic differentiation between the three species on a PC plot tells you anything about the existence of potential episodes of introgression. To distinguish between recent episodes of introgression and balanced TSP, I would rather look at the length of the shared regions.
- L124-126: Did you keep all 157+15+14 lines for the PC plot? If yes, I would randomly subsample within the *D magna* species, as PCA are strongly sensitive to such uneven sample sizes (see e.g., McVean, 2009, PLoS Genetics, e1000686).
- L146-148: Did you built the phylogenetic trees including the focal sites? If yes, this analysis needs to be redone as the same information cannot be used twice.
- L154-157: Why hypothesize that TSP fall in coding genes and thus restrict your analyses to annotated genes? Furthermore, the variance of this ratio is influenced by gene length, so you need to account for that. I would rather calculate the PtoD ratio in sliding windows of variable but pre-defined length across the genome.
- L217-227: Mostly out of curiosity, I am wondering whether you could have used a filter based on coverage (more stringent than the one you used excluding sites outside of 10-100) to exclude regions with probable mismapping issues, and whether this would have detected the problematic region on contig 52F, so that you could have filtered out these regions earlier in your analyses.
- L238-239: "Overall, we found a significant overrepresentation of TSPs in and surrounding of disease-related loci." Is this claim backed up by an enrichment statistical analysis?
- Did you correct your p-values for multiple testing (notably the wilcoxon rank sum test)?

Minor comments:

- L80 (and throughout the text): "Daphnia genotypes were tested for their resistance" I found the sentence unclear at first as, to me, the term "genotype" usually refers to one locus. I would suggest replacing occurrences of "genotypes" by clones or lines, for more clarity. Also, for readers not familiar with *Daphnia*, maybe state in the main text that the collected *Daphnia* individuals were maintained as clones in lab conditions?
- L84-86: I would add the same numbers for *D magna*, for comparison. Furthermore, I would include the result about the differential susceptibility between *D sinensis* and *D similis* in the main instead of the Method section ("D. sinensis was found to be, in general, more susceptible to *Pasteuria* (48 %, 47 out of 98 attachment tests) than *D. similis* (13 %, 14 out of 105 tests, Table S1)."), I would also add that number for *D magna* for comparison, and I would add a statistical test about these differences.
- Methods L57-58: please revise the sentence.
- L128-130: I am surprised you see the geographic differentiation of *D magna* on the PC plots fig S1-S2, but not on Figure 1, why would that be?
- L130: FigS1-S3: dots and legend too small to properly see the color
- Fig S3: Why do you see 2 groups for *D similis*? Does it correspond to any known factor?

- L146-148: Would/could you have recombination rate information to locally calculate the expected length of the genomic region with signals of balancing selection?
- L157-160: Please detail how d (the number of fixed SNPs between species) is computed (to which node / using which species pair)?
- Methods L168-169: did you phase separately the different species/populations, or all jointly?
- L231-232: a verb is missing.

Reviewer #2:

Remarks to the Author:

Review of Cornetti et al., "Long-term balancing selection..."

Submission 434467 to Nature Communications

July 2021

(Sorry for delaying receipt of your reviews, authors. I took a long time to write this. I'm sorry about that, to you and the journal office).

This paper by Cornetti et al. argues for evidence of a rare trans-species polymorphism (TSP). We know that mechanisms like heterozygote advantage and negative frequency dependent selection (NFDS) can maintain genetic (allelic diversity), preventing purging of alleles by 'purifying selection. If I understand correctly, it is especially interesting – and rare – to find such an ancient one as a TSP. A TSP is shared among related species, meaning that is very likely was inherited by a common ancestor. The authors make the argument for such a TSP using several lines of evidence.

But first, before the data: they have a fantastic and extremely well studied system in which to show it. Using a zooplankton host (three *Daphnia* species) and a bacterial parasite, they have several key ingredients to find it. First, they have the enemy, a highly virulent (often castrating) parasite. It is known to have a genetic specificity mechanism of infection that leads to NFDS; there are known alleles of genes that can confer resistance to it. The authors have developed the background, therefore, of decades of work to make an argument like this. (Plus, the host reproduces clonally, making the work here, while very laborious, very special). [Below, I wonder if the authors couldn't sell the logic and power of the system just a bit more?]

To make the argument,

(1) The authors collect a whole lot of *Daphnia* from basically all over. They use sequence data to argue that they in fact are distinct. That is important to rule out evidence for hybridization, etc., creating a TSP-like signature without TSP. (fig 1)

(2) They establish this nasty bacteria can infect clonal genotypes of all three species using a test they have developed. (This could be explained a bit more – it is crucial that the parasite can infect all three species still because ___[I'm not sure as explained, I think so we can argue it can still serve a selective force in nature, generating NFDS]) (fig 2)

(3) They show that alleles sequenced in key candidate regions group more closely to each other than the species do, etc. (fig 3) and that these candidates show statistically significant evidence for polymorphism (table 1)

(4) The candidates discovered 'make sense' from a resistance-perspective to this parasite (table 1).

I found it all to be well-done and convincing. There are many, many pages of supporting figures to help bolster the argument.

Most of my feedback below aims at work on making the paper more reader-friendly. I think the

authors often make presumptions about background of readers that I would not feel comfortable making. Perhaps my suggestions will help readers better follow and connect to the argument. I offer then with best of intention.

(1) Line 35: Just a consideration – perhaps broad readership would benefit from a definition of ‘balancing selection’. Then, and more importantly, on line 44, I think broader readership would benefit from understanding, right way, why they are special and important. Stated perhaps flippantly, ‘who cares that they are ancient’? The intro paragraph kind of assumes that all readers appreciate why polymorphisms inherited from a common ancestor and then maintained in isolated lineages is so cool. That is too bad! (Sorry if this is obvious to the authors, but I’d suggest maybe it isn’t obvious to all – and ‘rare’ does not mean ‘interesting’ or ‘important’ or ‘changing of thinking’ like we want to see in these magazines). Would it be particularly profound for NFDS in a RQ dynamic to produce a TSP? Or is a RQ system the most likely one to see such a TPS, since disease loci seem to provide some of the best evidence of TSP already? I left the first paragraph wanting a more concrete set up concerning these bigger picture issues, written for educated people who may not necessarily be in the know. All of that set up and explanation would help convince readers that the results are profound at e.g., line 278.

(2). Line 64 – I wish the explanation of this genomic signature could be more complete. It seems like important set up, but without reading the paper (ref 14), one can’t discern much. What supergene? What kind of signature? Can it be explained pithily?

(3). Line 74: please forgive me if I missed it, but I never really understood how the resistance phenotype information was used beyond qualitatively. Yes, I understand it was measured, but how did it inform the argument? I mean, I must underestimate that it was crucial to show that all species can become infected by those strains? The text here isn’t explaining what is special about the strains, it is skipping over genetic specificity issues of infection in this system, etc. (I know the *D. magna*-*Pasteuria* system is famous, thanks to the authors, but as a result, I’m finding here, up learn line 61, to ready too generally, not quite precisely enough. Perhaps related: I see the nice pictures in figure 2. How about translating some of the data in the Table in the Appendix to make a point beyond a picture but also beyond text? (In sum – the authors may think I am overthinking it, but I would explain clearly why it is crucial to show that a couple of strains of a *D. magna*-infecting *Pasteuria* can infect the others, because not-infecting would mean _____, but other less successful strains would mean _____. The authors can disagree, feeling like they don’t have to explain the obvious so obviously).

(4). Line 81: One of the worst parts of magazine articles is that things are often not explained clearly. Here is an example in this manuscript. What is a ‘parasite attachment test’ (without making readers have to dig up ref 21? And without sending us to the methods – something simple, but easy to understand, right here and now?)

(5). Line 128+: I was not sure why we have so many PCA plots. Can the authors think of statistical tests for multivariate groupings to use instead of showing me all of the various plots? Is there something I’m missing about the strategy that could be explained?

Then, figure S1 – 4, also 5:

- Caption – could you please describe this a bit more? PCA of what? Etc.
- Please definite WE, EA, and NA in the caption.
- I wish the ticks and axis labels were a bit bigger and the dots, too, so I could differentiate the colors more readily.
- I think that I missed why we use all four different PCAs instead of just one with a more formal analysis? What is gained by doing this PCA so many different ways? I think all along the way, broad readers would like some take-home messages explained. “In a PCA, recent introgression would look like _____, but our results are so distinct that we can rule that out”.

- What is *D. hispanica*? Could you describe it?

(6). Line 136-139: I found this long sentence to read awkwardly. Ditto: line 177-180 (which is very unfortunately, because it is crucial that all broad readers can 'see' and understand the authors' points here). Lines 223-227 (and I wonder if we need all of this detail). Line 246-249 (mainly because this explanation becomes very jargon-laden – can't this be said more crisply and simply, here and in the next sentence or two?).

(7). Line 148+: I lost track of why we see data separated into five lineages, then re-pooled back into 3 (for species-level analysis). Was it to find more candidate regions? Can the authors explain simply?

(8). Table S1:

- What is C1? C19? P20? In general, as I'm about to say, I found captions of figures and tables in esp. the Appendix to read unnecessarily and unhelpfully pithily.

(9). Line 151-3, and Tables S5-S6: This point applies to most tables and figures: since there is such a premium on space, as your reader I'd love it if there were more explanation in captions and more cross talk between figures, tables, captions, and text. For instance, tables S5 and S6 have columns of base pair lengths that I think I just read about (above) but were not explained. We get no help from the captions.

(10). Line 154: I think the authors imagine broad readership understand the logic here of PtoD better than I feel comfortable expecting. In short: the authors don't really explain, and they should. (The idea isn't really explain, but details about taking into account mutation rate are snuck in, which doesn't help with the explanation).

(11). Figure 3 and Table 1:

- Lines 172-3: Could the authors link these to 'transcript' in Table 1 to aid in cross-talk?

- It would be awesome if the candidates could be presented, in the various places they occur, in parallel order. It just makes it easier for us to read (and tightens presentation).

- It is confusing how the NACT-domain containing protein results are lumped in Table 1 but separate in Figure 3 – and they are out of order in Figure 3 (panels b and e are lumped). This is not easy to follow.

- It would be very helpful if Table 1 could explain 'gene function' a bit more, and how it relates to the *Daphnia*-*Pasteuria* system or could. I see some of that info in the text. But imagine a re-arranged Table 1, focused mutually on *** results of significance and giving readers a one-stop-shop to learn about this functional part of the argument? I would have appreciated it.

(12). Line 230+: I think broad readership would benefit from a bit more text about 'ABC supergene cluster' and 'F resistance locus' without making us read ref 12 and 27, respectively. I think your readers would appreciate more packaging and more help here. (If the authors had to move some details about specifics to appendix to make room for such packaging, I'd support that).

(13). Line 243: If I were the authors, I would rewrite this paragraph. It seems like a crucial part of the argument – the putative TSPs identified 'make sense' in that they are close to / at genes known to matter. But here, the authors use too many words (in my opinion) and get too details about mechanisms. I'd simplify explanations, write with fewer words (too bulky of expression), and remember to constantly look for ways to convince readers that this makes sense for *Pasteuria*. Just an example – line 259 mentions bacteria recognition. I would not be shocked if broad readership forgot *Pasteuria* was a bacteria. But also, once again, anyone not family if "F locus" is lost (and maybe everyone does know it already, and I'm wrong). There is so much explanation of e.g., ankryin, details extending way beyond the inference from the actual assays and data and bioinformatics here.

(14). Line 274+: I like this step-back to evaluate the results in a broader context, and a reminder of

the awesome *Daphnia*-*Pasteuria* system. I could see shrinking this some to make room for sketching out the nature of the argument. These magazine formats are essentially structureless. As a result, I think readers here would appreciate a sum-up of 'how' the argument was made. I did a little bit of this in my appraisal at the top, but I know the authors could do it much better than I did.

Reviewer #3:

Remarks to the Author:

This study uses extensive genomic sequencing of host clones from three closely related species and equally extensive lab assays of susceptibility to a virulent bacterial parasite to identify trans-species polymorphisms in a well-studied planktonic crustacean. The three host species in the genus *Daphnia* are all fairly closely related but also clearly genetically distinct. They can all be infected by the same bacterial parasite, *Pasteuria ramosa*, as shown in this study. After demonstrating that particular strains of the parasite can infect some but not all of the clones within each of the three host species, the study uses genomic analyses to identify five loci that show the fingerprints of balancing selection. Three of the five genes are in or near loci previously identified as being involved in resistance to this parasite (in studies in just one of the species in the present study, *Daphnia magna*). Overall, this is a very impressive amount of work. The more genomic aspects of the study are outside my area of expertise, but my overall opinion is that the work seems to be very carefully done overall and represents a very strong contribution to the literature.

The main thing I was left hoping for after reading the study is whether it would be possible to more directly link the results of the lab assays of attachment and infection with the genotypes at the identified loci. Do any of the alleles correspond with the outcomes of the attachment assays or infection assays? The study represents a very large amount of work, so I hesitate to ask for additional analyses, but it would be very interesting if it was possible to link specific alleles with infection outcomes. If it is not possible to do so, it would be helpful to briefly explain why in the study, as I suspect other readers would also hope for that link.

In a similar vein, but potentially of interest primarily to a narrower set of readers, is that it would be nice to have more of a link with some of the prior work that is primarily discussed in the supplement. On lines 57-61, they note that prior work has found attachment polymorphisms across *Daphnia* species when exposed to *Pasteuria*, but also that, if the *Pasteuria* strain came from a different species of *Daphnia*, infections always failed later in the infection process, perhaps because the prior studies used more divergent host species. In the present study, all of the *Pasteuria* strains were originally isolated from *D. magna*; of those that could attach to the other two species, there were successful infections in many of the host clones. What is unclear to me after reading the paper is if the authors hypothesize that the TSP identified are controlling attachment and/or later stages in the infection process (if it's possible to do the analysis suggested in the previous paragraph, that might answer this)? Moreover, would they expect to see similar alleles in other species (e.g., *D. pulex* clones that were used in the prior Luijckx et al. study)?

Specific comments:

1. Lines 48-49: Is the phrase "disease-related genes to evolve under" necessary in this sentence? I think the sentence is true without it and, if so, that it would be stronger without that phrase.
2. Lines 80-86: I know space is limited in this sort of paper, but it would help to give more of a summary of the results of the attachment and infection assays. Right now, the main text says that, in *D. magna*, resistance polymorphisms are widespread across the entire species, and that "similar polymorphisms" were seen for the other two species. However, no summary statistics are given that cover all three of the species, and no statistical test is used to support that statement. Table S1 gives information for each host genotype about whether they were susceptible or resistant to attachment by each of the five strains of *Pasteuria*, but it is hard to pull out the large patterns from it. Table S2 gives information about attachment and also about whether genotypes in which attachment occurred

developed full infections, but it does not include any data for magna. Some things I wondered included:

a) Why isn't magna included in table S2?

b) Was there a difference in the average number of strains that were able to attach across the three species? That is, if you calculated the # of strains with "S" for each of the rows in Table 1 and then averaged that within a species, how similar/different would the resulting three numbers be? In the case of *D. magna*, it would also be interesting to calculate this for each of the three subgroups.

c) For each of the *Pasteuria* strains, was there a difference in the proportion of clones of a given species that they were able to attach to? In this case, it would be summarizing the number of "S"s in a given column of Table S1, grouping by host species, and then comparing. (Again, it would also be interesting to see the numbers for each of the three groups of magna.)

d) Lines 84-85 give the proportions of *D. similis* and *D. sinensis* that were attachment positive that then became fully infected. What is the proportion for *D. magna*? Lines 72-73 in the supplement suggest it might be 100% but it's not clear whether that applies to the clones used in this study, since it cites an earlier study. (Related: The *similis* and *sinensis* result is in both the main text and supplement.)

e) Similarly, in the supplement, on lines 53-56, it gives the specific proportions for *similis* and *sinensis* susceptibility attachment but not magna.

3. The images in Figure 2 are very nice, but, in my opinion, are not particularly necessary in the main text. Using that space for some of the additions above would be preferable, in my opinion.

4. Line 180: It would help to give the reader more guidance on which specific parts of the figure support the pattern, such as by adding something like "(e.g., when a circle shows all three colors, as at the top of panel a)".

5. Lines 279-280: As written, it's not clear whether the phrase "we found only few instances of identical-by-descent TSPs" is referring to in the literature or in the present study. I think it's the latter, but it would help to specify.

6. PCA figures: Is there a reason why PCA1 is always plotted on the y-axis? In my experience, PCA1 is plotted on the x-axis.

7. Resistotype assays from the supplement: A) why was the hindgut assessed for only two of the five strains? B) How variable would the results be if you used different criteria for "susceptible" or "resistant" (e.g., susceptible if at least one host had attachment)?

8. Supplement lines 338-340: It would help to follow up on this idea more. What are those features?

Point-by-point response to reviewers.

Please find below a point-by-point response to the issues raised by the reviewers. Our text is in blue. The references to the literature mentioned below in abbreviated form, are all part of the manuscript can be found in full length there.

In summary, this is what we did:

We took great care about the technical aspects raised by reviewer #1. This somewhat more stringent analysis resulted in a reduction (from 5 to 4) of the final number of regions under long-term balancing selection. We also fixed the points suggested by reviewer #2 to make the text flow better and be clearer. The suggestion of reviewer #3 to test for associations between specific alleles and infection outcome is unfortunately not feasible. The reason is that the known resistance loci are all epistatically linked, i.e. the effect of one allele at a locus is contingent on the genotypes at other resistance loci. Thus, presence / absence of single alleles does not associate strongly with resistance. In large datasets with a less complex genetic background (only one population, instead of a worldwide sample) one can correct for these epistatic interactions when the genetic background is known (see Dexter et al 2023, MBE, who did this with a large sample from one population), but the worldwide sample used here is too diverse for this approach.

We use now the longer Nature Communication format. This longer format allowed us to explain many points better and in more detail.

With best wishes,

dieter ebert

REVIEWER COMMENTS

Reviewer #1 (Remarks to the Author):

The manuscript by Cornetti et al aims to detect genomic regions under long-term balancing selection, by looking for trans-species polymorphisms shared between three closely related crustacean species (*Daphnia magna*, *D. similis* and *D. sinensis*). They identified 14 genes containing identical-by-descent SNPs, of which 5 further exhibit evidence for balancing selection, and three are near loci involved in resistance to a known pathogen (*Pasteuria ramose*).

I found the manuscript of great interest and took a lot of pleasure reading it, as it combines thorough population genomics analyses with experimental work to decipher the evolutionary dynamics of loci under antagonistic coevolution. Furthermore, the text and figures are clear and well interpreted.

Thank you very much for the encouraging words about our work.

Main comments:

- Could you elaborate more in the main text on why the resistance/susceptible phenotype is so

variable across *Pasteuria* strains (Table S1, Methods L50-56)? Is there a significant interaction between the *Pasteuria* strain and the host species/population?

This is an important point. Indeed, the host and the parasite have a very specific form of genetic interactions. We elaborate now on this point in the introduction. The new format gives us the necessary space to do so.

- L122-124: I don't think that observing an overall genetic differentiation between the three species on a PC plot tells you anything about the existence of potential episodes of introgression. To distinguish between recent episodes of introgression and balanced TSP, I would rather look at the length of the shared regions.

We have now included a direct assessment of genome wide signals of introgression in the form of an ABBA-BABA test. This assessment suggests that there is no evidence for introgression between the three focal species.

- L124-126: Did you keep all 157+15+14 lines for the PC plot? If yes, I would randomly subsample within the *D. magna* species, as PCA are strongly sensitive to such uneven sample sizes (see e.g., McVean, 2009, PLoS Genetics, e1000686).

Good point. We randomly subsampled the clones of *D. magna* to 15 samples and re-run the PCA several times. The result is exactly the same as before. We added 9 such plots to the supplement to illustrate this point (see Fig. S1).

- L146-148: Did you build the phylogenetic trees including the focal sites? If yes, this analysis needs to be redone as the same information cannot be used twice.

In the previous version, we did not exclude the focal sites. We now repeated the analyses excluding the focal site in the sequences used for building the phylogenetic trees. As partly expected, the probability of obtaining allelic trees slightly decrease, especially for short fragments, but we still have very clear outliers that do not contradict the previous results. Because of this the list of putative TSPs reported now in the revised version is one less than what we previously had identified. Due to a higher variation among species for the PtoD estimates, we reduced the cut-off value from 90 to 85 %.

- L154-157: Why hypothesize that TSP fall in coding genes and thus restrict your analyses to annotated genes? Furthermore, the variance of this ratio is influenced by gene length, so you need to account for that. I would rather calculate the PtoD ratio in sliding windows of variable but pre-defined length across the genome.

We did not do a sliding window approach because of the very wide spread structural polymorphisms in the genome, in particular, in the regions around the resistance genes, which are structurally extremely diverse (Bento et al 2017, PLoS Genetics; Ameline et al 2020, MBE; Fredericksen et al 2023, PLoS Genetics). Furthermore, all our genomes are assembled from short reads, and therefore resulted in assemblies of highly variable quality, in particular in non-coding regions (they often have lower complexity). The combination of short-read

assemblies and high structural variation among host genotypes made the sliding window approach not feasible.

Following the comment regarding gene-length, we now did the analysis in which PtoD is corrected by gene length. Indeed, we observed a slightly positive correlation ($R=0.06$, $P<0.05$) between PtoD and gene length. To correct for this, we divided the calculated PtoD for the gene length and multiplied this value by 1000 (for making it more readable). Our best candidates were considered the ones falling into the top 10 % distribution for the three species simultaneously. One candidate gene observed earlier (on contig 18F) did not fully satisfy this filter (borderline as 15% in *D. magna*, <10% in the other two species). However, since this gene is in close proximity of a known resistance locus, we kept it conditionally in the list of potential candidate genes.

- L217-227: Mostly out of curiosity, I am wondering whether you could have used a filter based on coverage (more stringent than the one you used excluding sites outside of 10-100) to exclude regions with probable mismapping issues, and whether this would have detected the problematic region on contig 52F, so that you could have filtered out these regions earlier in your analyses.

We played with different coverages and it did not make a difference. In particular for the region on contig 52F we were unable to improve. However, contig 52F dropped out of the list of candidate TSP after the revised the analysis following reviewer 1 suggestion.

- L238-239: “Overall, we found a significant overrepresentation of TSPs in and surrounding of disease-related loci.” Is this claim backed up by an enrichment statistical analysis?

We did play around with an enrichment analysis, but did not include it in the manuscript. We feel uncomfortable with the assumptions that go into such an analysis: Do you base it on SNPs, genes, sliding windows, contigs, ...)? What distance between the resistance loci and the focal TSP is still acceptable? And other assumptions. Therefore, we considered an enrichment analysis not constructive and did not report it.

To avoid the issue in the revised manuscript, we avoid the claim that it is a “significant overrepresentation”.

- Did you correct your p-values for multiple testing (notably the wilcoxon rank sum test)?

We had not done this in the earlier version, but did it now. We use the FDR to do this. It hardly changes anything.

Minor comments:

- L80 (and throughout the text): “Daphnia genotypes were tested for their resistance” I found the sentence unclear at first as, to me, the term “genotype” usually refers to one locus. I would suggest replacing occurrences of “genotypes” by clones or lines, for more clarity. Also, for readers not familiar with Daphnia, maybe state in the main text that the collected Daphnia individuals were maintained as clones in lab conditions?

The sentence “Daphnia genotypes were tested for their resistance” reads now “Daphnia clones were tested for their resistance to five strains of *P. ramosa* isolated from *D. magna* populations across Europe, using the parasite attachment test”.

We explain now explicitly that Daphnia were maintained as clones under lab conditions. Furthermore, we use now "clone" instead of "genotype" for the Daphnia lines used in the study. We use "genotype" only in cases we refer to loci.

- L84-86: I would add the same numbers for *D. magna*, for comparison. Furthermore, I would include the result about the differential susceptibility between *D. sinensis* and *D. similis* in the main instead of the Method section (“*D. sinensis* was found to be, in general, more susceptible to *Pasteuria* (48 %, 47 out of 98 attachment tests) than *D. similis* (13 %, 14 out of 105 tests, Table S1).”), I would also add that number for *D. magna* for comparison, and I would add a statistical test about these differences.

We added now more information about the attachment tests and the infection tests to the main result section. We include a new table summarizing the results of all attachment tests of all three species and also for the three clades of *daphnia magna*. This table shows well the differences among the species.

- Methods L57-58: please revise the sentence.

We revised this sentence, but also edited parts of the previous paragraph to make the entire section clearer.

- L128-130: I am surprised you see the geographic differentiation of *D. magna* on the PC plots fig S1-S2, but not on Figure 1, why would that be?

This is because the difference between *D. magna* and the other species is so big, that the three *D. magna* clades are not visible when all three species are included. When we include only 2 species, we see the clades of *D. magna* already in PC2, otherwise it hides in a higher order PC.

- L130: FigS1-S3: dots and legend too small to properly see the color

We changed this.

- Fig S3: Why do you see 2 groups for *D. similis*? Does it correspond to any known factor?

This is because we sampled this species around its type-locality in Israel, and then in Russia. In between Russia and Israel we were not able to obtain samples. Therefore, the two groups represent the two geographic sampling regions.

- L146-148: Would/could you have recombination rate information to locally calculate the expected length of the genomic region with signals of balancing selection?

We have a recombination map of *D. magna*, but it is not fine grained enough to make a reasonable estimate here.

- L157-160: Please detail how d (the number of fixed SNPs between species) is computed (to which node / using which species pair)?

This explanation was accidentally forgotten. We explain it now.

- Methods L168-169: did you phase separately the different species/populations, or all jointly?

We phased all samples at once, including the use of individual BAM files in order to incorporate read-based phasing. This is now mentioned in the text.

- L231-232: a verb is missing.

Fixed.

Reviewer #2 (Remarks to the Author):

Review of Cornetti et al., “Long-term balancing selection...”
Submission 434467 to Nature Communications
July 2021

(Sorry for delaying receipt of your reviews, authors. I took a long time to write this. I’m sorry about that, to you and the journal office).

This paper by Cornetti et al. argues for evidence of a rare trans-species polymorphism (TSP). We know that mechanisms like heterozygote advantage and negative frequency dependent selection (NFDS) can maintain genetic (allelic diversity), preventing purging of alleles by ‘purifying selection. If I understand correctly, it is especially interesting – and rare – to find such an ancient one as a TSP. A TSP is shared among related species, meaning that is very likely was inherited by a common ancestor. The authors make the argument for such a TSP using several lines of evidence.

But first, before the data: they have a fantastic and extremely well studied system in which to show it. Using a zooplankton host (three *Daphnia* species) and a bacterial parasite, they have several key ingredients to find it. First, they have the enemy, a highly virulent (often castrating) parasite. It is known to have a genetic specificity mechanism of infection that leads to NFDS; there are known alleles of genes that can confer resistance to it. The authors have developed the background, therefore, of decades of work to make an argument like this. (Plus, the host reproduces clonally, making the work here, while very laborious, very special). [Below, I wonder if the authors couldn’t sell the logic and power of the system just a bit more?]

Thank you for the nice words. We worked hard to make the logic of the project and the power of the system clearer in the revised text.

To make the argument,

(1) The authors collect a whole lot of *Daphnia* from basically all over. They use sequence data to argue that they in fact are distinct. That is important to rule out evidence for hybridization, etc., creating a TSP-like signature without TSP. (fig 1)

Also here is asked for a test of hybridization/introgression. (See above comments by reviewer 1). In our revision we have now included a direct assessment of genome wide signals of introgression in the form of an ABBA-BABA test. This assessment suggests that there is no introgression between the three focal species.

(2) They establish this nasty bacteria can infect clonal genotypes of all three species using a test they have developed. (This could be explained a bit more – it is crucial that the parasite can infect all three species still because ___[I'm not sure as explained, I think so we can argue it can still serve a selective force in nature, generating NFDS]) (fig 2)

We revised the sections related to this and made it clearer.

(3) They show that alleles sequenced in key candidate regions group more closely to each other than the species do, etc. (fig 3) and that these candidates show statistically significant evidence for polymorphism (table 1).

Yes.

(4) The candidates discovered ‘make sense’ from a resistance-perspective to this parasite (table 1).

I found it all to be well-done and convincing. There are many, many pages of supporting figures to help bolster the argument.

Most of my feedback below aims at work on making the paper more reader-friendly. I think the authors often make presumptions about background of readers that I would not feel comfortable making. Perhaps my suggestions will help readers better follow and connect to the argument. I offer then with best of intention.

We followed these advices and revised the text at many places to make the text more reader-friendly.

(1) Line 35: Just a consideration – perhaps broad readership would benefit from a definition of ‘balancing selection’. Then, and more importantly, on line 44, I think broader readership would benefit from understanding, right way, why they are special and important. Stated perhaps flippantly, ‘who cares that they are ancient’? The intro paragraph kind of assumes that all readers appreciate why polymorphisms inherited from a common ancestor and then maintained in isolated lineages is so cool. That is too bad! (Sorry if this is obvious to the authors, but I’d suggest maybe it isn’t obvious to all – and ‘rare’ does not mean ‘interesting’ or ‘important’ or ‘changing of thinking’ like we want to see in these magazines). Would it be particularly profound for NFDS in a RQ dynamic to produce TSP? Or is a RQ system the most likely one to see such a TPS, since disease loci seem to provide some of the best evidence of TSP already? I left the first paragraph wanting a more concrete set up concerning these bigger picture issues, written for educated people who may not necessarily be in the

know. All of that set up and explanation would help convince readers that the results are profound at e.g., line 278.

We define balancing selection more clearly and make clear why it is of interest to study it in general and in the context of transspecies polymorphism and infectious diseases. We removed the term NFDS, as it is a related concept, but not the same as balancing selection. The text works better without it. I hope the revised first paragraph brings our message better across. TSP are rare, but can have any of various reasons (e.g. sex chromosomes, self-incompatibility loci). Outside these well understood cases, the Red Queen theory provides the best explanation for TSP at disease related loci.

(2). Line 64 – I wish the explanation of this genomic signature could be more complete. It seems like important set up, but without reading the paper (ref 14), one can't discern much. What supergene? What kind of signature? Can it be explained pithily?

We elaborate on this now and make it clearer. The sentence reads now: "In a Eurasian sample of genomes from *D. magna* it has been shown that, at a group of *P. ramosa* resistance genes, forming a supergene, balancing selection can be inferred from the patterns of genomic variation observed."

(3). Line 74: please forgive me if I missed it, but I never really understood how the resistance phenotype information was used beyond qualitatively. Yes, I understand it was measured, but how did it inform the argument? I mean, I must underestimate that it was crucial to show that all species can become infected by those strains? The text here isn't explaining what is special about the strains, it is skipping over genetic specificity issues of infection in this system, etc. (I know the *D. magna*-*Pasteuria* system is famous, thanks to the authors, but as a result, I'm finding here, up learn line 61, to ready too generally, not quite precisely enough. Perhaps related: I see the nice pictures in figure 2. How about translating some of the data in the Table in the Appendix to make a point beyond a picture but also beyond text? (In sum – the authors may think I am overthinking it, but I would explain clearly why it is crucial to show that a couple of strains of a *D. magna*-infecting *Pasteuria* can infect the others, because not-infecting would mean ____, but other less successful strains would mean _____. The authors can disagree, feeling like they don't have to explain the obvious so obviously).

This point relates to our suggestion, that we know the parasite that caused coevolution across deep time. Therefore, it is crucial to show that all three host species show highly variable response to the different parasite isolates. We followed the suggestion to produce a new table that shows the infection data for the three species and for each parasite isolate. All this information is now much better incorporated into the flow of the logic.

(4). Line 81: One of the worst parts of magazine articles is that things are often not explained clearly. Here is an example in this manuscript. What is a 'parasite attachment test' (without making readers have to dig up ref 21? And without sending us to the methods – something simple, but easy to understand, right here and now?)

We explain this test now when we introduce it first at the beginning of the result section: "Daphnia clones were tested for their resistance to five strains of *P. ramosa* isolated from *D.*

magna populations across Europe. For this we used the parasite attachment test, a test conducted with fluorescent parasite spores, which can be observed in attachment-positive individuals adhering to the fore- or hindgut wall."

(5). Line 128+: I was not sure why we have so many PCA plots. Can the authors think of statistical tests for multivariate groupings to use instead of showing me all of the various plots? Is there something I'm missing about the strategy that could be explained?

PCAs are complex and provide a lot of information. It is possible to reduce this information, but specialists like to see the PCAs in details to judge the results. This is why we provide these details. But we place them in the supplement, so the average reader is not bothered by it.

Then, figure S1 – 4, also 5:

- Caption – could you please describe this a bit more? PCA of what? Etc.
- Please definite WE, EA, and NA in the caption.
- I wish the ticks and axis labels were a bit bigger and the dots, too, so I could differentiate the colors more readily.
- I think that I missed why we use all four different PCAs instead of just one with a more formal analysis? What is gained by doing this PCA so many different ways? I think all along the way, broad readers would like some take-home messages explained. "In a PCA, recent introgression would look like _____, but our results are so distinct that we can rule that out".

We added the missing information to the caption and increased the size of the dots and the fonts and ticks on the axes.

Regarding the different PCAs and as stated already before, PCAs provide complex output with a lot of information. It is possible to reduce this information, but specialists like to see the PCAs in details to judge the results. This is why we provide these details. But we place them in the supplement, so the average reader is not bothered by it.

- What is *D. hispanica*? Could you describe it?

D. hispanica is another *Daphnia* species in the "D. magna group" of species. It is closely related to the three species used here, but not as close as the three focal species are to each other. Therefore, it was used as an outgroup to root the tree.

(6). Line 136-139: I found this long sentence to read awkwardly. Ditto: line 177-180 (which is very unfortunately, because it is crucial that all broad readers can 'see' and understand the authors' points here). Lines 223-227 (and I wonder if we need all of this detail). Line 246-249 (mainly because this explanation becomes very jargon-laden – can't this be said more crisply and simply, here and in the next sentence or two?).

We revised these 4 places in the main text to make them clear (adding more information or condensing the information).

(7). Line 148+: I lost track of why we see data separated into five lineages, then re-pooled back into 3 (for species-level analysis). Was it to find more candidate regions? Can the authors explain simply?

No, this was not done to "find more candidate regions", but to be more stringent. We started the analysis with all five clades (three species and three clades within *D. magna*). This provided us with a list of candidate SNPs that were polymorphic in all five clades. This is a more stringent criterium than using already here only the three species clades. It excludes SNPs where one allele is found in one *D. magna* clade and the other in another clade, which may happen because of the geographic spread of the clades across three continents. Pooling the *D. magna* data from the beginning would have resulted in more candidate SNPs, but many of them would be clade specific and would thus not be informative for our analysis. Our approach results in a candidate list where each polymorphisms is observed in all five clades. We explain this in the text now.

(8). Table S1:

- What is C1? C19? P20? In general, as I'm about to say, I found captions of figures and tables in esp. the Appendix to read unnecessarily and unhelpfully pithily.

C1, C19 and P20 are the labels for the isolates of the *Pasteuria* parasite. These are just names. We explain this now. More generally, we tried to make all figure and table legends more readable.

(9). Line 151-3, and Tables S5-S6: This point applies to most tables and figures: since there is such a premium on space, as your reader I'd love it if there were more explanation in captions and more cross talk between figures, tables, captions, and text. For instance, tables S5 and S6 have columns of base pair lengths that I think I just read about (above) but were not explained. We get no help from the captions.

We made all figure and table legends more readable.

(10). Line 154: I think the authors imagine broad readership understand the logic here of PtoD better than I feel comfortable expecting. In short: the authors don't really explain, and they should. (The idea isn't really explain, but details about taking into account mutation rate are snuck in, which doesn't help with the explanation).

We revised the text and added further explanation here to make this clear. The change of text layout (from Nature to Nat. Comm.) made this possible.

(11). Figure 3 and Table 1:

- Lines 172-3: Could the authors link these to 'transcript' in Table 1 to aid in cross-talk?
- It would be awesome if the candidates could be presented, in the various places they occur, in parallel order. It just makes it easier for us to read (and tightens presentation).
- It is confusing how the NACT-domain containing protein results are lumped in Table 1 but separate in Figure 3 – and they are out of order in Figure 3 (panels b and e are lumped). This

is not easy to follow.

We rearranged the layout of the figure to bring the trees belonging to closely located sites closer to each other. This aids also the link to table 1, as the trees are presented now in the same order as in table 1. More generally, we present the candidates now everywhere in the same order.

- It would be very helpful if Table 1 could explain ‘gene function’ a bit more, and how it relates to the Daphnia-Pasteuria system or could. I see some of that info in the text. But image a re-arranged Table 1, focused mutually on *** results of significance and giving readers a one-stop-shop to learn about this functional part of the argument? I would have appreciated it.

The function of Table 1 is to present the statical evidence supporting our hypothesis. It is not aimed to add biological information here. We added the very short description of the function of the genes, to make the link between the table and the text (where all these functions are discussed and explained) stronger. Mixing the statistical aspects with more biological aspects would be odd. We therefore do not follow the advice to add more biological information on gene function to this table.

(12). Line 230+: I think broad readership would benefit from a bit more text about ‘ABC supergene cluster’ and ‘F resistance locus’ without making us read ref 12 and 27, respectively. I think your readers would appreciate more packaging and more help here. (If the authors had to move some details about specifics to appendix to make room for such packaging, I’d support that).

We revised this section and made it clear.

(13). Line 243: If I were the authors, I would rewrite this paragraph. It seems like a crucial part of the argument – the putative TSPs identified ‘make sense’ in that they are close to / at genes known to matter. But here, the authors use too many words (in my opinion) and get too details about mechanisms. I’d simplify explanations, write with fewer words (too bulky of expression), and remember to constantly look for ways to convince readers that this makes sense for Pasteuria. Just an example – line 259 mentions bacteria recognition. I would not be shocked if broad readership forgot Pasteuria was a bacteria. But also, once again, anyone not family if “F locus” is lost (and maybe everyone does know it already, and I’m wrong). There is so much explanation of e.g., ankryin, details extending way beyond the inference from the actual assays and data and bioinformatics here.

We do agree that this paragraph is dense and complex. But the information given here is clearly need in this manuscript. This manuscript would not fly without a detailed analysis of the possible role of the candidate genes. We packed all of this in one paragraph towards the end of the discussion to allow the readers not interested in these details to skip over it. Shortening the paragraph would make it less clear and so we left it mostly as we think it brings the information needed here across.

(14). Line 274+: I like this step-back to evaluate the results in a broader context, and a reminder of the awesome Daphnia-Pasteuria system. I could see shrinking this some to make room for sketching out the nature of the argument. These magazine formats are essentially structureless. As a result, I think readers here would appreciate a sum-up of ‘how’ the argument was made. I did a little bit of this in my appraisal at the top, but I know the authors could do it much better than I did.

With the new format of the manuscript we decided to raise this paragraph to the status of a "Conclusion". But then we read that Nature Communication does not allow Conclusions. So we produced a paragraph starting with " In conclusion, our study provides evidence ... ". We worked on it to make it stronger following the suggestion given here.

Reviewer #3 (Remarks to the Author):

This study uses extensive genomic sequencing of host clones from three closely related species and equally extensive lab assays of susceptibility to a virulent bacterial parasite to identify trans-species polymorphisms in a well-studied planktonic crustacean. The three host species in the genus *Daphnia* are all fairly closely related but also clearly genetically distinct. They can all be infected by the same bacterial parasite, *Pasteuria ramosa*, as shown in this study. After demonstrating that particular strains of the parasite can infect some but not all of the clones within each of the three host species, the study uses genomic analyses to identify five loci that show the fingerprints of balancing selection. Three of the five genes are in or near loci previously identified as being involved in resistance to this parasite (in studies in just one of the species in the present study, *Daphnia magna*). Overall, this is a very impressive amount of work. The more genomic aspects of the study are outside my area of expertise, but my overall opinion is that the work seems to be very carefully done overall and represents a very strong contribution to the literature.

Thank you very much for this encouraging words.

The main thing I was left hoping for after reading the study is whether it would be possible to more directly link the results of the lab assays of attachment and infection with the genotypes at the identified loci. Do any of the alleles correspond with the outcomes of the attachment assays or infection assays? The study represents a very large amount of work, so I hesitate to ask for additional analyses, but it would be very interesting if it was possible to link specific alleles with infection outcomes. If it is not possible to do so, it would be helpful to briefly explain why in the study, as I suspect other readers would also hope for that link.

This is indeed a pressing question. However, to test for associations between specific alleles and infection outcome is unfortunately not feasible. The reason is that resistance loci are all epistatically linked, i.e. the effect of one allele at a locus is contingent on the genotypes at other resistance loci. Thus, presence / absence of alleles at one locus does not associate strongly with resistance. In large datasets with a less complex genetic background (e.g. only one population, instead of a worldwide sample) one can correct for these epistatic interactions when the genetic background is known (see Dexter et al 2023, MBE did this with a large sample from one population), but the worldwide sample used here has a too high level of genetic complexity for this approach.

In a similar vein, but potentially of interest primarily to a narrower set of readers, is that it would be nice to have more of a link with some of the prior work that is primarily discussed in the supplement. On lines 57-61, they note that prior work has found attachment polymorphisms across *Daphnia* species when exposed to *Pasteuria*, but also that, if the *Pasteuria* strain came from a different species of *Daphnia*, infections always failed later in the infection process, perhaps because the prior studies used more divergent host species. In the present study, all of the *Pasteuria* strains were originally isolated from *D. magna*; of those that could attach to the other two species, there were successful infections in many of the host clones. What is unclear to me after reading the paper is if the authors hypothesize that the TSP identified are controlling attachment and/or later stages in the infection process (if it's possible to do the analysis suggested in the previous paragraph, that might answer this)? Moreover, would they expect to see similar alleles in other species (e.g., *D. pulex* clones that were used in the prior Luijckx et al. study)?

We revised this section intensively (it had to be short, because the ms was originally submitted to Nature). The new text explains many of these details in greater length and better. Attachment is the main polymorphism observed in all our studies. It is a very strong predictor of host resistance/susceptibility. Therefore, we believe the pattern observed here are also due to attachment. We make this now clear in the revised manuscript.

Specific comments:

1. Lines 48-49: Is the phrase “disease-related genes to evolve under” necessary in this sentence? I think the sentence is true without it and, if so, that it would be stronger without that phrase.

We agree and followed this suggestion.

2. Lines 80-86: I know space is limited in this sort of paper, but it would help to give more of a summary of the results of the attachment and infection assays. Right now, the main text says that, in *D. magna*, resistance polymorphisms are widespread across the entire species, and that “similar polymorphisms” were seen for the other two species. However, no summary statistics are given that cover all three of the species, and no statistical test is used to support that statement. Table S1 gives information for each host genotype about whether they were susceptible or resistant to attachment by each of the five strains of *Pasteuria*, but it is hard to pull out the large patterns from it. Table S2 gives information about attachment and also about whether genotypes in which attachment occurred developed full infections, but it does not include any data for *magna*.

This corresponds with a comment by one of the other reviewers, asking for more information here. We introduce a new table (Table 1) that summarizes the results of the attachment tests with the three species and three clades of *D. magna* and rewrote this entire paragraph to be more clear on this.

Some things I wondered included:

a) Why isn't *magna* included in table S2?

This has been published for *Daphnia magna* already earlier. We explain this now in the revised manuscript and refer to the earlier papers.

b) Was there a difference in the average number of strains that were able to attach across the three species? That is, if you calculated the # of strains with “S” for each of the rows in Table 1 and then averaged that within a species, how similar/different would the resulting three numbers be? In the case of *D. magna*, it would also be interesting to calculate this for each of the three subgroups.

Yes, there is a difference. The new Table 1 has this information. It was however highly variable among clones within species, ranging from total resistance (all R) to nearly total susceptibility (mostly S). This is known for a long time. Clones which are all R, are still susceptible to other *Pasteuria* isolates. We never found a clone that was universally resistant.

c) For each of the *Pasteuria* strains, was there a difference in the proportion of clones of a given species that they were able to attach to? In this case, it would be summarizing the number of “S”s in a given column of Table S1, grouping by host species, and then comparing. (Again, it would also be interesting to see the numbers for each of the three groups of *magna*.)

Yes, different *Pasteuria* isolates differ in their overall infectivity. This is in part explained by the attachment site (hindgut attachment is more common (40 - 60 %) than foregut attachment (15-25%)), but also depends on the host species. The new Table 1 has this information.

d) Lines 84-85 give the proportions of *D. similis* and *D. sinensis* that were attachment positive that then became fully infected. What is the proportion for *D. magna*? Lines 72-73 in the supplement suggest it might be 100% but it’s not clear whether that applies to the clones used in this study, since it cites an earlier study. (Related: The *similis* and *sinensis* result is in both the main text and supplement.)

For *D. magna* this proportion is 100%. This was already published earlier. We explain this now.

e) Similarly, in the supplement, on lines 53-56, it gives the specific proportions for *similis* and *sinensis* susceptibility attachment but not *magna*.

We fixed this.

3. The images in Figure 2 are very nice, but, in my opinion, are not particularly necessary in the main text. Using that space for some of the additions above would be preferable, in my opinion.

We believe it is important to show that all three species can become infected and that they show the same infection phenotype. This is why we think this figure is helpful.

4. Line 180: It would help to give the reader more guidance on which specific parts of the figure support the pattern, such as by adding something like “(e.g., when a circle shows all three colors, as at the top of panel a)”.

We did this now.

5. Lines 279-280: As written, it’s not clear whether the phrase “we found only few instances of identical-by-descent TSPs” is referring to in the literature or in the present study. I think it’s the latter, but it would help to specify.

We did this now.

6. PCA figures: Is there a reason why PCA1 is always plotted on the y-axis? In my experience, PCA1 is plotted on the x-axis.

This was a neglect of us. We changed the axes in the figures in the supplement.

7. Resistotype assays from the supplement: A) why was the hindgut assessed for only two of the five strains?

Only these two strains use the hindgut for attachment. C1, C19 and P20 never attach to the hindgut. We explain this now.

B) How variable would the results be if you used different criteria for “susceptible” or “resistant” (e.g., susceptible if at least one host had attachment)?

Using "susceptible if at least one host had attachment" would bring the average infectivity for all parasite isolates up and average resistance down. This effect would be very small for C1 and C19 and would be moderate for P20. For P15 and P21 it would be a little bit higher, as hindgut attachment shows more stochastic variation (Bento et al 2020, Heredity). The overall results of our analysis would not be impacted by this.

8. Supplement lines 338-340: It would help to follow up on this idea more. What are those features?

We were not able to locate this place. Our supplement does not have line numbers and in the main text the line number given here are in the references section. A search under "features" did not reveal places that seemed likely to be right. So we cannot comment here.

Reviewers' Comments:

Reviewer #1:

Remarks to the Author:

The authors have thoroughly addressed the comments I had on the previous version of their manuscript and nicely incorporated the associated revisions in this new version.

Notably:

- i) they have added a test (ABBA-BABA) to show the absence of introgression between their three focal crustacean species (though I am unsure why the value for *D. sinensis* <-> *D. similis* is not provided);
- ii) they reran the PCA by evenly subsampling their clones and showed it did not affect their structuration;
- iii) they obtained phylogenetic trees without the focal site to verify the signal still holds (which is the case in 4 out of their 5 previous outliers);
- iv) they corrected their PtoD ratio for gene length (by the way, for the one with 15% in *D. magna* and 3-5% in the other two species, could it be that the selective pressures differ in intensity or timing between the *magna* and *sinensis/similis* lineages ?);
- v) they included correction for multiple testing.

As for the functional enrichment analysis, I agree there are multiple ways to do it, but I would have reported the different results when changing the parameters for more transparency. The simplest approach might be to assign each TSP to the nearest gene (whatever the distance) and see whether there is an enrichment in this list compared to the genomic list (but that is somewhat arbitrary of course). Anyway, the authors have rewritten their sentence without mentioning a significance, so that is fair.

As for the calculation of *d* (the number of fixed differences between species), the value for *sinensis* and *similis* are not independent as it is currently calculated. I am also unclear how the SNPs with a *Fst* of 1 in the *D. magna* vs *D. sinensis* comparison for *sinensis* can be different from those with a *Fst* of 1 in the *D. sinensis* vs *D. magna* comparison for *magna*...? I would suggest to reconstruct the state of the ancestral node *sinensis/similis* (as it is the only node you can reconstruct anyway given you don't have an outgroup, right ?) and establish the number of fixed SNP of each species to this node, so these PtoD ratios can represent independent information.

Overall, I find this manuscript very interesting and well suited for Nature Communications.

Reviewer #2:

None

Reviewer #3:

Remarks to the Author:

This is a revised version of a manuscript focused on balancing selection at loci related to pathogen resistance, with a particular focus on trans-species polymorphisms. The authors conclude the abstract by saying "These findings support the theory that specific antagonistic coevolution is able to maintain genetic diversity over millions of years". I agree with the authors that their findings provide support for that, and that this is a very interesting finding.

The authors have thoroughly addressed all of the comments raised in the first round of review. I look forward to seeing this article in print.

There was one comment of mine from the first round of review where the authors noted uncertainty over the line number reference, which left them unable to respond to the comment in particular. My comment was about the sentence that, in the original submission, read "This gene is a member of the

large “Cladoceran-specific protein” family and has several features suggesting an involvement in the interaction between *Daphnia* and *Pasteuria*.” That was where the paragraph ended in the original submission. The revision has rewritten this section in a way that addresses my question. Thus, while they weren’t sure of what it referred to, the revision addresses the original concern.

Point-by-point response to reviewers.

Please find below a point-by-point response to the issues raised by the reviewers. Our text is in blue. The references to the literature mentioned below in abbreviated form, are all part of the manuscript can be found in full length there.

With best wishes,

dieter ebert

REVIEWER COMMENTS

Reviewer #1 (Remarks to the Author):

The authors have thoroughly addressed the comments I had on the previous version of their manuscript and nicely incorporated the associated revisions in this new version.

Notably:

- i) they have added a test (ABBA-BABA) to show the absence of introgression between their three focal crustacean species (though I am unsure why the value for *D. sinensis* <-> *D. similis* is not provided);
- ii) they reran the PCA by evenly subsampling their clones and showed it did not affect their structuration;
- iii) they obtained phylogenetic trees without the focal site to verify the signal still holds (which is the case in 4 out of their 5 previous outliers);
- iv) they corrected their PtoD ratio for gene length (by the way, for the one with 15% in *D. magna* and 3-5%% in the other two species, could it be that the selective pressures differ in intensity or timing between the magna and sinensis/similis lineages ?);
- v) they included correction for multiple testing.

As for the functional enrichment analysis, I agree there are multiple ways to do it, but I would have reported the different results when changing the parameters for more transparency. The simplest approach might be to assign each TSP to the nearest gene (whatever the distance) and see whether there is an enrichment in this list compared to the genomic list (but that is somewhat arbitrary of course). Anyway, the authors have rewritten their sentence without mentioning a significance, so that is fair.

The suggestion made by the referee would be ok for a genomic region with normal recombination (i.e. an approximately even recombination rate across the genome). This is however not the case here. All but one of the so far discovered resistance genes sit in regions of highly reduced recombination. Thus, neighbouring genes are not as independent in their evolution as one would wish for. Any meaningful enrichment analysis would require independence of the focal genes and the gene(s) one would like to treat as a "control group". We therefore feel not comfortable position to employ such an analysis here.

As for the calculation of d (the number of fixed differences between species), the value for *D. sinensis* and *D. similis* are not independent as it is currently calculated. I am also unclear how the SNPs with a F_{st} of 1 in the *D. magna* vs *D. sinensis* comparison for *D. sinensis* can be different from those with a F_{st} of 1 in the *D. sinensis* vs *D. magna* comparison for *D. magna*...? I would suggest to reconstruct the state of the ancestral node *D. sinensis/similis* (as it is the only node you can reconstruct anyway given you don't have an outgroup, right ?) and establish the number of fixed SNP of each species to this node, so these PtoD ratios can represent independent information.

Before going into the details of this comment, we want to stress that the focus of our study is to understand if the polymorphisms we observe in our focal species *D. magna* are shared with other species, i.e. to test if variation observed in *D. magna* is shared with variation in the sister taxon. The sister taxon includes multiple species. Given the limited material available to us we included two species (*D. sinensis* and *D. similis*) within the sister taxon, as this would strengthen our analysis. However, the contrast of these two species (*D. sinensis* and *D. similis*) is not our focus and given the small sample size, is not expected to be rewarding. Furthermore, for these two species, we have much less knowledge about their interactions with the parasite (*Pasteuria ramosa*), as compared to *D. magna*. We therefore placed less emphasis on *D. sinensis* and *D. similis*, as it would distract from the main focus of the study, namely the polymorphisms maintained through time from the deepest node in our three species assembly. This is the common procedure in studies of similar nature as our study, namely to focus on the contrast related to the deepest node. For example, the focus of Teixeira et al (2015, MBE) was humans in contrast to the sister taxon. Likewise, the study of König et al. (2019, eLife) using three species within the plant genus *Capsella*. The aim is to find evidence for TSP across the oldest node in the total sample.

This is also among the reasons why we have not included estimates of the ABBA-BABA test for introgression between *D. sinensis* and *D. similis*. Even if these species would show introgression (although PCA (Figure S3) and Admixture analyses (not shown here) strongly suggest it is not the case), it would not invalidate the finding of the contrast of *D. magna* with the sister taxon. Additionally, on a technical side, our current sample set fails to conform to a phylogenetic configuration which is robust to the basic assumptions of the original ABBA-BABA (i.e., an additional outgroup to the *D. sinensis* and *D. similis* node is needed).

For the first point ("As for the calculation of d ...") in the above comment by referee 1: We agree that the d value for *D. sinensis* and *D. similis* are not independent. This is because both are compared to *D. magna*. This is part of the process, however we believe it is not a problem, as we do not treat them as independent.

For the second point "I am also unclear how the SNPs with a F_{st} of 1 in the *D. magna*?" Yes, the number of fixed SNPs is not different in the mentioned comparisons (i.e. *D. magna* vs *D. sinensis* and *D. sinensis* vs *D. magna*) and this was not expected. What changes in the reported PtoD ratio is p (number of polymorphisms), that are different from species to species. In the original version of the manuscript for *D. magna* d calculation we had compared *D. magna* to *D. sinensis*. This was an arbitrary choice as we did not expect much difference with the other species (*D. similis*). Now we added the comparison to *D. similis* and indeed, the results are very close, with the results being highly correlated (correlation = 0.96). We added these data to Table 2 and S7 (see below for technical details). The observation that we obtain very similar statistics for the two species strongly strengthens our message, and is

in line with our general approach, namely testing for polymorphisms that are present in each of the three species. Since we have now two values (*D. magna* compared to two species) that are not truly independent, we changed the criteria for filtering our candidates. We now included genes in which the average percentile of all PtoD values is below 10. We end up with the same four genes, so nothing changed.

An analysis including the reconstructed state ("I would suggest to reconstruct the state of the ancestral node *sinensis/similis* ...") of the ancestral node *sinensis/similis* would not be meaningful in our eyes, as it would result in a PtoD statistic that would combine within and between species aspects. Furthermore, it would tell us something about the distribution of past polymorphisms, but we want to understand current polymorphisms. We therefore consider an analysis based on the reconstruction of the ancestral node *sinensis/similis* as not helpful within the frame of our current study. We do like however the idea in general and keep it in mind for our ongoing analysis of the patterns we observed in the deeper phylogeny of the entire family of the Daphniidae and related other Cladocera that we are currently working on.

Overall, I find this manuscript very interesting and well suited for Nature Communications.

Thank you very much.

Reviewer #3 (Remarks to the Author):

This is a revised version of a manuscript focused on balancing selection at loci related to pathogen resistance, with a particular focus on trans-species polymorphisms. The authors conclude the abstract by saying "These findings support the theory that specific antagonistic coevolution is able to maintain genetic diversity over millions of years". I agree with the authors that their findings provide support for that, and that this is a very interesting finding.

Thank you very much.

The authors have thoroughly addressed all of the comments raised in the first round of review. I look forward to seeing this article in print.

There was one comment of mine from the first round of review where the authors noted uncertainty over the line number reference, which left them unable to respond to the comment in particular. My comment was about the sentence that, in the original submission, read "This gene is a member of the large "Cladoceran-specific protein" family and has several features suggesting an involvement in the interaction between *Daphnia* and *Pasteuria*." That was where the paragraph ended in the original submission. The revision has rewritten this section in a way that addresses my question. Thus, while they weren't sure of what it referred to, the revision addresses the original concern.

OK.

Reviewers' Comments:

Reviewer #1:

Remarks to the Author:

The authors have addressed all of my comments. I have no further comments and I look forward to seeing this article published.